# On the Significance of Softmax Geometry: Interpretability and Token Decoding

## Abstract

Large language models represent their internal state as high-dimensional vectors. Many tasks of practical interest require measuring similarity between these vectors. Usually, this similarity is measured with a Euclidean notion. Recent work has argued that Euclidean geometry is ill-matched to semantic structure represented by LLMs. However, it's unclear whether this mismatch actually has practical consequences. In this paper, we study the practical effect of the similarity measure in the softmax layer of large language models (where the geometry is best understood). We consider two tasks: (1) learning interpretable features using sparse autoencoders, and (2) scalable retrieval of the top-$k$ most probable next tokens given a context. In both cases, we find that using the correct geometry dramatically improves the performance.

## 1 Introduction

Many tasks of practical interest in large language models (LLMs) require, either implicitly or explicitly, measuring how similar activation or weight vectors are to one another. Commonly, this vector similarity is measured using Euclidean distance (or cosine similarity), which is computationally convenient and easy to implement. However, the use of Euclidean geometry seems somewhat arbitrary. Indeed, it has been argued that the 'natural' geometry of LLMs is non-Euclidean (Park et al., 2024; 2025). This begs the question: can moving from Euclidean similarity to some other notion of similarity actually lead to improved performance on tasks of practical interest in LLMs?

A challenge here is that the internal geometry of LLMs is not well understood in general. Since we do not generally know the 'right' way to measure similarity in the first place it's not simple to determine how much improvement may be possible by moving away from Euclidean geometry. The main exception to this is the geometry of the final (softmax) layer, which is significantly better understood (Park et al., 2024; 2025). Then, the aim of this paper is to study the practical importance of non-Euclidean LLM geometry by understanding its effect on tasks in the final LLM layer.

We focus on two main tasks. The first is learning interpretable features using sparse autoencoders (SAEs), a subject that has attracted considerable attention in the LLM context (e.g., Cunningham et al., 2023; Bricken et al., 2023; Lieberum et al., 2024; Templeton et al., 2024; Gao et al., 2025). Training an autoencoder requires specifying a similarity measure to evaluate the quality of the reconstruction. The question is whether changing the similarity measure from Euclidean distance to a more appropriate measure can lead to more interpretable features. The second task we consider is scalable finding of the $k$ most probable next tokens given the context embedding (i.e., the tokens with highest probability under the softmax distribution). This may be viewed as a retrieval task in a vector database, where the unembedding vectors are the database items and the context embedding is the query. If the underlying geometry were Euclidean, this approximate top-$k$ problem could be solved with locality sensitive hashing (LSH) (Datar et al., 2004; Andoni & Indyk, 2008; Jafari et al., 2021). The question here is whether (and how) we can use the non-Euclidean geometry to do better? We note that beyond the link to vector databases, scalable softmax is of intrinsic interest in its own right—in particular, it allows a decoupling of the vocabulary size and computational cost of generation, which can be valuable as models continue to scale (Kaplan et al., 2020).

The main finding of this paper is that using the correct geometry does indeed have a dramatic effect on the performance of both tasks. In detail:

1. Park et al. (2024) give a similarity metric for unembedding vectors based on the geometry of the softmax function. We show that training sparse autoencoders for word unembeddings using this metric does indeed improve interpretability.

2. Next, we use the softmax geometry to develop a scalable algorithm for approximating the top-$k$ most probable tokens. This makes use of the sparse autoencoder learned in the previous step, and a particular non-linear *'dual map'* that sends context embeddings to the space where the unembedding vectors live.

3. Finally, we show that this approximate nearest neighbor scheme works well and, in particular, strongly outperforms (Euclidean) locality sensitive hashing.

These results suggest that there is significant headroom for improvement in practical tasks by moving from Euclidean geometry to one that is better adapted to the structure of large language models.

## 2 BACKGROUND

**LLM Basics**  We begin by fixing some notation. Consider the next token prediction task in LLMs, and let $x$ be a context sentence and $y \in \mathcal{Y}$ the next token sampled by the model. The LLM first encodes $x$ into an embedding vector $\lambda_x \in \Lambda \cong \mathbb{R}^d$, and assigns each token $y$ an unembedding vector $\gamma_y \in \Gamma \cong \mathbb{R}^d$. The conditional probability $p(y \mid x)$ is given by the softmax distribution:

$$p(y \mid x) = \frac{\exp\left(\lambda_x^\top \gamma_y\right)}{\sum_{y'} \exp\left(\lambda_x^\top \gamma_{y'}\right)}. \tag{2.1}$$

**Vector Similarity in LLMs**  Many tasks in the analysis and manipulation of LLMs require measuring how similar pairs of embedding or unembedding vectors (more generally, weight vectors) are to one another. Frequently, this is done using a Euclidean notion of similarity, such as the dot product or cosine similarity. However, it's unclear if, and when, this is the best choice.

Observe that the embedding vectors $\lambda_x$ and unembedding vectors $\gamma_y$ do not live in the same space. This is significant because the softmax distribution equation 2.1 *only* constrains the interaction between embedding and unembedding vectors (via the dot product $\lambda_x^\top \gamma_y$). Nothing in a LLM directly constrains interactions between pairs of embedding vectors, nor pairs of unembedding vectors. Specifically, if we define a map $\tilde{\gamma} = M\gamma$ and $\tilde{\lambda} = M^{-\top}\lambda$ for some invertible matrix $M$ then the bi-products are invariant—$\lambda^\top \gamma = \tilde{\lambda}^\top \tilde{\gamma}$—but dot products between embedding-embedding and unembedding-unembedding vectors are not—i.e., $\gamma_i^\top \gamma_j \neq \tilde{\gamma}_i^\top \tilde{\gamma}_j$ and $\lambda_i^\top \lambda_j \neq \tilde{\lambda}_i^\top \tilde{\lambda}_j$ for $i \neq j$. This means, in particular, that learning a model with a softmax distribution does not privilege Euclidean geometry.

Motivated by this, Park et al. (2024) argue (in particular) that Euclidean geometry does not align well with the structure of semantics as understood by humans. For example, the unembeddings corresponding to words that are semantically similar (e.g., "lion" and "tiger") need not be closer in a Euclidean sense than two words that are semantically dissimilar (e.g., "lion" and "car"), and similarly for the embedding vectors. This motivates the main question of this paper: can moving from Euclidean geometry to one that better reflects semantic structure lead to improved performance on tasks of practical interest in LLMs?

To answer this question, we require a candidate for a better geometry. In general, the geometric structure of the internal layers of LLMs remains poorly understood. However, Park et al. (2024) do pick out a particular choice of geometry on the unembedding space at the final layer that they argue is well aligned with semantic structure. Specifically, they show that using the particular inner product

$$\langle \gamma, \gamma' \rangle_C = \gamma^\top \mathsf{Cov}^{-1}\gamma', \quad \gamma, \gamma' \in \Gamma. \tag{2.2}$$

where Cov is the covariance matrix of unembeddings, aligns well with human intuitions about semantic similarity. This can viewed as first applying a whitening transformation to the unembedding vectors. They show that, if high-level concepts are represented as directions in the unembedding space (the so-called *linear representation hypothesis*), then representations of "causally separable" concepts (e.g., male-female and English-French) are orthogonal under this inner product (dubbed the *causal inner product*).

## 3 GEOMETRY-AWARE SPARSE AUTOENCODER FOR UNEMBEDDINGS

As shown by Park et al. (2024), the causal inner product reveals linear features in the unembedding space. In this section, we extend that insight by training a sparse autoencoder (Bricken et al., 2023) to evaluate whether this geometric approach can help revealing interpretable features.

**Sparse autoencoders** A large body of work has focused on using SAEs to learn dictionaries for interpreting the internal representations of LLMs (e.g., Cunningham et al., 2023; Bricken et al., 2023; Templeton et al., 2024; Lieberum et al., 2024; Gao et al., 2025). The general idea is to map the input data (typically the context embedding at a particular token position and layer) to a sparse representation in a latent space, and then to reconstruct the input data from this representation using a dictionary (decoder). That is,

$$\hat{\mathbf{x}} = \mathbf{D}(\sigma(\mathbf{E}\mathbf{x})) \tag{3.1}$$

where $\mathbf{x}$ is the input data, $\mathbf{E}$ is the encoder matrix, $\sigma$ is a non-linear activation function, and $\mathbf{D}$ is the decoder matrix. The hope is that $\mathbf{E}$ will contain interpretable dictionary atoms (features) that can be used to understand the model's behavior. Following Gao et al. (2025), we consider Top-K sparse autoencoders to induce sparsity. Effective dictionary learning requires a similarity measure to evaluate the quality of the reconstruction. The default assumption has been to use Euclidean distance (with the exception of Braun et al. (2024)), which raises the concerns discussed above. In this section, we validate this theoretical concern by training (i) an SAE on the unembedding matrix and (ii) an SAE on the unembedding matrix *under the transformation corresponding to the causal inner product*.

We focus on the unembedding matrix because its geometry is better understood (Park et al., 2024; 2025) than the geometry of the context embeddings at layers near the end (but not the final layer) of the model, which is where the SAE is typically applied (e.g., Gao et al., 2025).

**SAEs trained with the causal inner product produce more interpretable features.** We evaluate the quality of the SAE by examining a selection of words, extracting their latent codes, and then interpreting the corresponding features by printing the words whose latent codes most strongly activate the corresponding features. While the SAE trained using the unwhitened data produces a dictionary with some interpretable features, many of the features lack any coherent meaning. In contrast, the SAE trained using the causal inner product produces a dictionary with many more interpretable features. For example, in Figure S1, we show the top 5 features for the word "puppy" and the top 10 words that activate each feature. The SAE without whitening yields a clear *dog* feature, but the other features appear nonsensical. In contrast, the SAE with whitening yields all coherent features. Interestingly, we see that certain incoherent features from the unwhitened SAE (such as feature 2751) appear frequently in the latent codes of very unrelated words like "puppy" and "Bayesian". Additional comparisons on different words can be found in Appendix S1.2. For a quantitative evaluation, we rely on GPT-4o to judge the features produced by SAEs trained with and without whitening. The results in Figure 1 demonstrate that whitening leads to consistently better performance.

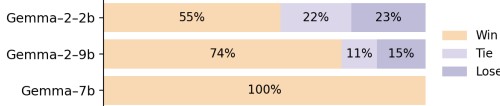

**Figure 1:** SAEs trained with the causal inner product yield more interpretable features. The figure reports the win rate, judged by GPT-4o, on the relevance of SAE features to target words, over a sample of 100 common words.

## 4 GEOMETRY-AWARE NEXT TOKEN PREDICTION

Next, we consider a retrieval task: finding the $k$ most probable next tokens given a context embedding. Precisely, given a context sentence $x$ the next token is selected according to the softmax distribution $p(y \mid x) \propto \exp(\lambda_x^\top \gamma_y)$, we're interested in finding the $k$ most probable tokens under this distribution. Of course, this could be done by simply computing the entire distribution and then sorting the tokens by their probabilities. However, in retrieval contexts we can often do much better than this naive approach. For example, locality sensitive hashing (LSH) is a well-known technique for finding

approximate nearest neighbors in a manner that avoids needing to compute most distances. The question we address in this section is whether we can leverage the softmax geometry to develop a more effective algorithm for the $k$ most probable tokens problem.

## 4.1 Locality Sensitive Hashing

We begin by recalling the LSH approach and explain how it can be applied in the context of the softmax problem. The high-level intuition of LSH is that if we can find a hash function that maps similar items to the same bucket then, given a new item, we can find its nearest neighbors by hashing it and only searching the items in the same bucket. We now turn to the details.

A hash function $H$ is called $(R, cR, p_1, p_2)$-sensitive if, for all $q, q'$ in $\mathbb{R}^d$, the following holds:

- if $\text{dist}(q, q') \leq R$, then $\mathbb{P}[H(q) = H(q')] \geq p_1$.
- if $\text{dist}(q, q') > cR$, then $\mathbb{P}[H(q) = H(q')] \leq p_2$.

where $\text{dist}(\cdot, \cdot)$ measures a generic distance measure between two points (Jafari et al., 2021). In the particular case when $\text{dist}(\cdot, \cdot)$ is the Euclidean distance, we can choose $H(p) = \left\lfloor \frac{a^\top p + b}{w} \right\rfloor$ where $a$ is a random $d$-dimensional vector chosen from a Gaussian distribution and $b$ is a real number uniformly sampled from $[0, w)$, where $w$ is the size of the hash bucket (Datar et al., 2004). When $\text{dist}(\cdot, \cdot)$ is the angular metric (Charikar, 2002), we can choose $H(p) = \text{sgn}(a^\top p)$ where $a$ is also a random $d$-dimensional vector drawn from a Gaussian distribution. In this case, if the angle between $p$ and $p'$ is $\theta$, then the probability they collide (i.e., $\mathbb{P}[H(q) = H(q')]$) is $1 - \theta/\pi$.

Given a hash function $H$, we can build a hash table by hashing all the items in the dataset. In practice, each hash table uses several locality-sensitive hash functions whose combined outputs define its buckets. Because these functions are randomized, we typically build multiple hash tables. We then employ a counting-based similarity measure: the relevance between a query (context embedding) and a key (token unembedding) is the number of hash tables in which they collide. Tokens with the highest counts form a candidate set, from which we compute softmax probabilities and select the top $k$ tokens.

**Choice of Hash Function** The key to the success of LSH is the choice of the hash function. In particular, the hash should align with whatever the relevant similarity measure is. In the case of the softmax problem, we are interested in $\lambda_x^\top \gamma_y$. Then, it may seem natural to use the angular metric—and the associated hash function—to solve the nearest neighbor problem. However, it is not clear whether this choice, which is inherently based on Euclidean geometry, is actually the best one. The question is whether we can improve performance by finding a hash function that aligns more closely with the softmax geometry.

## 4.2 Retrieval Geometry and the Duality Map

We are interested in the similarity between the context embedding $\lambda_x$ and the unembedding vectors $\{\gamma_y\}$. The challenge is that, as discussed in Section 2, the vectors $\lambda_x$ and $\gamma_y$ live in different spaces. Accordingly, measures like the cosine similarity between $\lambda_x$ and $\gamma_y$ need not be meaningful. In particular, this means that a hash that is well-suited for one space need not be well suited for the other. Figure 2 illustrates this point: applying the same projection to both embedding and unembedding vectors shows that they occupy wholly distinct regions. Then, for example, a hash that does a good job capturing similarity between unembeddings need not also do a good job capturing similarity between embeddings and unembeddings.

Now, if all the vectors lived in the space $\Gamma$, we would have a natural guess for a hash function. Namely, we could use the hash function corresponding to the angular metric *with respect to the causal inner product*. The problem is then how to hash the context embedding $\lambda_x$. The basic strategy we will use here is to first map the context embedding $\lambda_x \in \Lambda$ to a new vector $\gamma_x \in \Gamma$, then apply the hash function to $\gamma_x$. This requires us to find a suitable mapping from $\Lambda$ to $\Gamma$.

**Duality Mapping** We need a map that is compatible with the structure of softmax. To that end, we write the softmax distribution as

$$\mathbb{P}(y \mid x) = \exp\left(\lambda_x^\top \gamma_y - A(\lambda_x)\right), \tag{4.1}$$

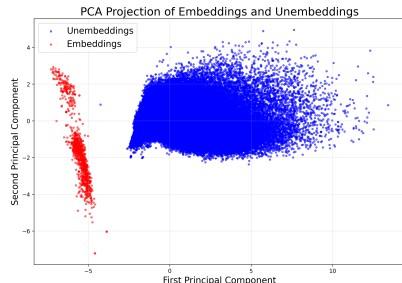

**Figure 2:** The embedding and unembedding vectors occupy distinct subspaces. The plots show PCA projection of embedding and unembedding vectors from Gemma-1 7B. The embedding vectors are sampled from 1000 random sentences.

where $A(\lambda) = \log \sum_{y \in \mathcal{Y}} \exp(\lambda^\top \gamma_y)$ is the log-normalizer (a convex function). We can now recognize this as an "exponential family" distribution with natural parameter $\lambda_x$. Exponential family distributions are a class of probability models that admit a wide range of special properties (Wainwright et al., 2008). For our purposes, the main importance is that the natural geometric structure of exponential family distributions is well-studied (Efron, 1978). In exponential family geometry, the relationship between the natural parameter space and its dual plays a prominent role. The idea here is to view the softmax as an exponential family distribution and then make use of the induced duality map from $\lambda_x$ into $\Gamma$.

For exponential families, the privileged dual map $\varphi$ is given as $\varphi(\lambda) := \nabla A(\lambda)$ (see, e.g., Wainwright et al., 2008; Efron, 2022). This is in some sense the unique map that is consistent with the interpretation of $\lambda$ as the parameter of a probability distribution. In our context, this map is:

$$\varphi(\lambda) := \nabla A(\lambda) = \mathbb{E}[\gamma_y \mid \lambda] = \sum_{y \in \mathcal{Y}} \frac{\exp(\lambda^\top \gamma_y)}{\sum_{y' \in \mathcal{Y}} \exp(\lambda^\top \gamma_{y'})} \gamma_y. \tag{4.2}$$

where the conditional expectation is with respect to the output token distribution induced by $\lambda$. We refer to this map as the *dual map*. (We note this map is onto the convex hull of the unembedding vectors $\{\gamma_y\}_{y \in \mathcal{Y}}$, which is contained in $\Gamma$).

**The Dual Hash**   Conceptually, our strategy is to define a hash function on the unembedding space using the causal inner product, then extend this hash function to the embedding vectors $\lambda \in \Lambda$ by computing the hash of the dual vector $\varphi(\lambda) \in \Gamma$. We could directly compute the dual map by explicitly computing the word probabilities and then taking the weighted average of the unembedding vectors. However, this would defeat the purpose of efficient softmax. Instead, we train a small neural network to approximate the dual map directly. We find this works well in practice, see Section 5.

### 4.3 SPARSE AUTOENCODER AS A HASH

We still need to define a hash function on the unembedding space. One obvious choice is to use a standard LSH construction with the causal inner product geometry. Another choice can be arrived at as follows: Suppose we have a large number of hash tables, each with one hash function. (As shown later in Figure 3, increasing the number of hash tables enhances performance, whereas adding more hash functions may degrade performance.) Recall that hash functions take the form of $\mathrm{sgn}(a^\top p)$ where $a$ is a random Gaussian vector and $p$ is the query vector. Let $a_i$ be the random vectors of the $i$-th hash table, we can write the assignment to hash tables as: $\mathrm{sgn}(A^\top p)$ where the $i$-th column of $A$ is $a_i$ where $\mathrm{sgn}$ applies elementwise. This effectively serves as the encoder of an autoencoder when the non-linear activation is the sign function.

Following this logic, we can interpret the sparse codes produced by the SAE as *learned* hash codes for the unembeddings (contrary to the random codes of the standard construction). Furthermore, when using ReLU as the activation function, the magnitudes of these sparse codes offer additional insight into the unembeddings (see Appendix S1.2). Ideally, the learned structure will lead to more effective hashes, where hash collisions reliably reflect semantic closeness. In Section 5 we find that indeed the SAE hash does a better job of preserving semantic structure.

**Summary** To summarize, to approximately retrieve the top-$K$ most probable tokens:

1. Map the context embedding into the whitened unembedding space using a dual map learned by a small neural network.
2. Pass the dual representation of the context embedding through a sparse autoencoder to generate a sparse code, then sort its nonzero entries by magnitude.
3. Collect tokens from the precomputed dictionary that share the same nonzero entry, starting with the largest entry; stop once the candidate set reaches its maximum size, breaking ties arbitrarily.

Note that once we have the candidate set, we can assign a probability to each element by directly computing the softmax on this set. This is just the original softmax distribution conditional on the output belonging to the candidate set. If the candidate set contains most of the probability mass, this will gives a highly accurate approximation to the original softmax distribution.

**On scalability** Let the full vocabulary be $V$ and the candidate subset $\tilde{V}$. This approximation reduces the time complexity from $O(|V|d)$ to $O(|\tilde{V}|d)$. Empirically (see Section 5.2), the size of $\tilde{V}$ is typically a tiny fraction of $V$, making this approach especially advantageous for very large vocabularies. Moreover, since greedy decoding or sampling generally only considers the highest-probability tokens—and most tokens have probabilities near zero—this approximation causes only a negligible loss of information (see Section 5.3 for results on text generation).

## 5 EXPERIMENTS

In this section, we present empirical results showing the effectiveness of geometry aware softmax for next token prediction. Here, the results are conducted on Gemma-1 7B (Team et al., 2024a) and Gemma-2 2B and 9B models (Team et al., 2024b). They share the same vocabulary, and Gemma-2 models are trained via distillation, enabling direct comparison across model sizes and training techniques under the same vocabulary.

Throughout the experiments, different methods would propose a candidate set, from which softmax probabilities are estimated and top-$k$ tokens are chosen. To measure the effectiveness of such proposed candidate set $\tilde{V}(x) \subseteq V$, given context sentence $x$, we use the following probability mass covering metric:

$$P(x) = \sum_{y \in \tilde{V}(x)} \mathbb{P}(y|x)$$

In other words, it calculates the proportion of the conditional next-token probability mass covered by the proposed candidate set; higher values indicate better coverage.

### 5.1 LOCALITY SENSITIVE HASHING AND DUAL MAP

We begin by evaluating the effectiveness of locality-sensitive hashing (LSH) in retrieving the most probable tokens. Sentences are sampled from the BookCorpus dataset (Zhu et al., 2015), with the first half of each sentence used as a context for next-token prediction.

As shown in Table 1, the vanilla LSH method underperforms out of the box. Inspired by Park et al. (2024), we whiten the unembedding vectors and apply the inverse whitening to the context embedding to align their subspaces. Although this adjustment yields a modest boost, performance remains subpar. Finally, we try applying LSH to unembeddings and exact dual representations of embeddings, which yields significant improvements over the baselines.

Additional experimental results across different parameter settings and model variants are available in Appendix S2. Gemma-2 models, in particular, demonstrate even stronger performance with the LSH method paired with the dual map. This is due to their distillation-based training, which yields high-confidence next-token predictions with many conditional probabilities approaching 1.

In practice, computing the exact dual maps requires computing the exact probabilities over the whole vocabulary, which defeats the purpose of hashing here. Therefore, instead of computing the exact dual map, we train a multi-layer perceptron with hidden dimension 2048 to learn a approximate dual map.

**Table 1:** Locality sensitive hashing performs best with dual map. The table reports the mean probability mass coverage ($\pm$ standard error) over 1,000 samples using the Gemma-1 7B model.

| (# Tables, # Functions) | LSH | LSH w/ Whitening | LSH w/ Exact Dual Map |
|---|---|---|---|
| $(16, 4)$ | $0.143 \pm 0.005$ | $0.118 \pm 0.004$ | $0.258 \pm 0.008$ |
| $(32, 4)$ | $0.067 \pm 0.004$ | $0.107 \pm 0.004$ | $0.510 \pm 0.009$ |
| $(64, 8)$ | $0.073 \pm 0.005$ | $0.223 \pm 0.006$ | $0.522 \pm 0.010$ |
| $(128, 8)$ | $0.093 \pm 0.005$ | $0.180 \pm 0.005$ | $0.570 \pm 0.010$ |

We collected the training data by extracting the first 256 tokens (using the Gemma tokenizer) from over 300,000 Wikipedia articles. From each 256-token sequence, we randomly sampled ten positions to retrieve the final-layer token embeddings and their corresponding dual representations, yielding more than three million training pairs in total. Here, the dual representations are whitened. As shown in Figure 3, with the approximate dual map, increasing the number of hash tables boosts probability mass coverage, but altering the number of hash functions per table can degrade performance.

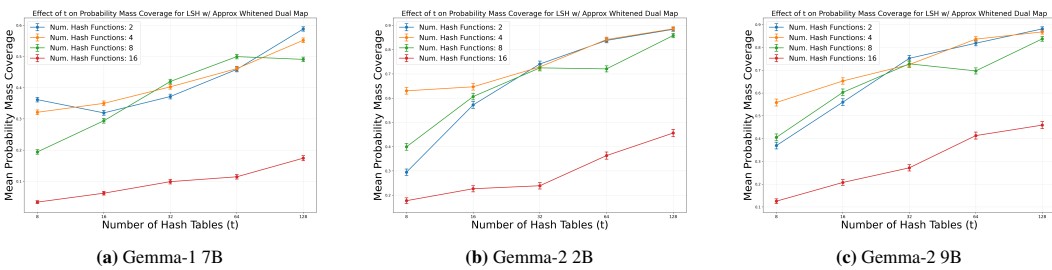

**(a)** Gemma-1 7B          **(b)** Gemma-2 2B          **(c)** Gemma-2 9B

**Figure 3:** Increasing the number of hash tables can improve performance. Figures show the effect of the number of hash tables on mean probability mass coverage for LSH with approximate whitened dual map on varying numbers of hash functions per table. Error bars indicate standard error over 1000 samples. Panels (a), (b) and (c) show results for the Gemma-1 7B, Gemma-2 2B and Gemma-2 9B models, respectively.

## 5.2 Sparse Autoencoder

In this section, we present results on using sparse autoecnoder to create hash for most probable token predictions. First of all, recall that we use sparse codes of the whitened dual representation of the context embeddings to select the candidate set. Suppose there is no limit on the size of the candidate set; Figure 4 shows that the top candidates often only accounts for less than 2% of the vocabulary size.

As shown in Figure 5, The SAE-based method outperforms the LSH-based method. As a reminder, LSH with whitening applies the causal inner product by whitening the unembedding space and then using the inverse whitening transform on the embeddings. For a fair comparison, we set the maximum candidate threshold to 8,000 for the Gemma-1 7B model and 5,000 for the Gemma-2 models. Note that because Gemma-2 models produce more sharply peaked next-token probabilities, their probability mass coverage greatly exceeds that of the Gemma-1 model. We set the number of hash tables to 128 and the number of hash functions per table to 2, as this configuration yields the best LSH performance. Additional experiments on different dataset can be found in Appendix S2 which shows the same pattern. On the other hand, Table 2 demonstrates that the SAE-based model retrieves top tokens more effectively. Additionally, as shown in Table S1 and Table S2, even for the Gemma-2B models—where LSH with exact dual functions excels at finding the top-1 token—SAE with approximate dual functions outperforms at retrieving the remaining top tokens.

## 5.3 Text Generation Judged by GPT-4

We generated 100 full sentences on diverse topics using GPT-4o, each 10–200 words long. The initial prompt is shown in Figure S16. Each sentence's first half is fed into the model to evaluate the quality of different text-generation methods. We test both greedy decoding and the sampling strategy with top $k$ ($k = 50$) and top $p$ ($p = 0.9$). Sampling-based generation is tested at sampling temperatures of 0.7 and 1.0. We evaluate both the model's default generation and generation using our SAE-based

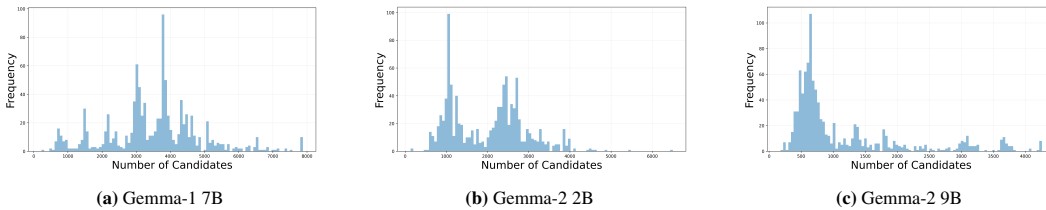

**(a)** Gemma-1 7B   **(b)** Gemma-2 2B   **(c)** Gemma-2 9B

**Figure 4:** The SAE-selected candidate set represents only a small fraction of the full vocabulary. The figure displays a histogram of candidate counts across different context sentences, based on 1,000 samples.

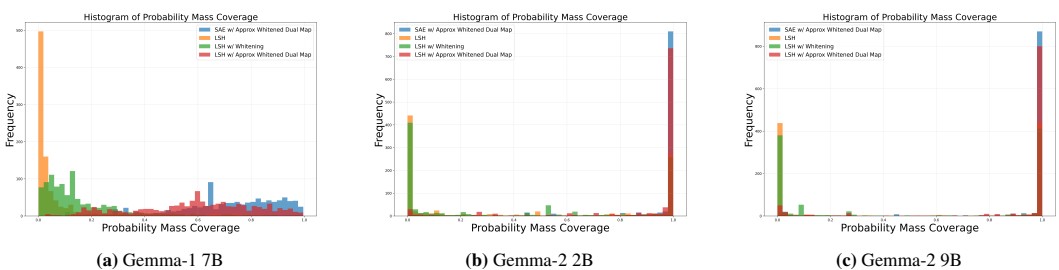

**(a)** Gemma-1 7B   **(b)** Gemma-2 2B   **(c)** Gemma-2 9B

**Figure 5:** The SAE-based approach consistently outperforms LSH-based methods. The figures depict histograms of probability mass convergence over 1000 samples from the BookCorpus dataset.

softmax. For both decoding methods, we cap new tokens at 100, 200, or 300 (see examples in Appendix S2.1). Outputs from the SAE method are fluent and human-like, so we systematically evaluate them by having GPT-4o judge default and SAE-based generations using the system prompt in Figure S17. The win/tie/lose rates in Figure 6 and Figure S18 show that SAE-based generation performs on par with the vanilla method generation.

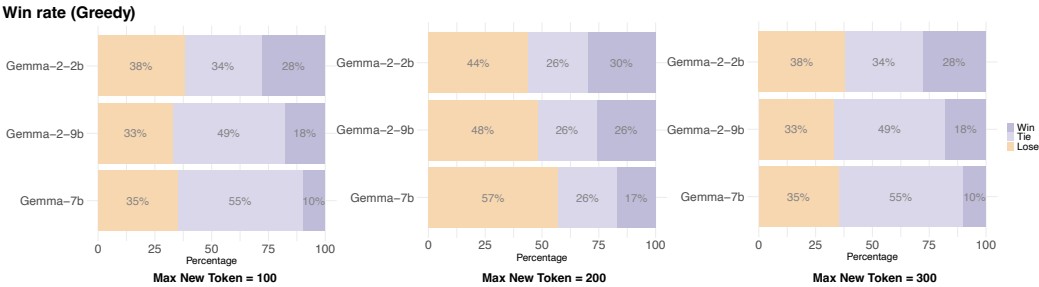

**Figure 6:** The SAE-based approach performs on par with default greedy decoding for sentence generation. The graphs show win rates as judged by GPT-4o.

# 6 RELATED WORK

**Representations of Semantic Structure in Language Models** This paper builds on a long line of work studying representations of semantical structures in language models, including those in word embedding spaces (e.g., Mikolov et al., 2013; Levy & Goldberg, 2014; Arora et al., 2016; Mimno & Thompson, 2017; Nickel & Kiela, 2017; Hewitt & Manning, 2019; Chiang et al., 2020) and in language model embedding spaces (e.g., Reif et al., 2019; Chen et al., 2021; Chang et al., 2022; Belinkov, 2022; Li et al., 2023; Jiang et al., 2023; Gurnee et al., 2024). A principal focus in this line of work is on *linear* representations, based on the idea that high-level semantic concepts are often represented as directions or vectors in an embedding space (e.g., Mikolov et al., 2013; Arora et al., 2016; Elhage et al., 2022; Park et al., 2024; Jiang et al., 2024). In particular, recent work studying the final softmax layer (Park et al., 2024; 2025) finds that linear representations

**Table 2:** The SAE-based approach outperforms LSH-based methods in retrieving the top-K tokens. The table shows the average recall rates for these top-K tokens, with standard errors computed over 1,000 samples using Gemma-1 7B model.

| Top-K | SAE w/ Approx. Dual | LSH w/ Exact Dual | LSH w/ Approx. Dual |
|-------|---------------------|-------------------|---------------------|
| 1     | $0.903 \pm 0.009$   | $0.670 \pm 0.015$ | $0.903 \pm 0.009$   |
| 5     | $4.147 \pm 0.033$   | $2.445 \pm 0.055$ | $3.689 \pm 0.040$   |
| 10    | $7.920 \pm 0.064$   | $4.212 \pm 0.102$ | $6.523 \pm 0.070$   |
| 20    | $14.83 \pm 0.125$   | $7.070 \pm 0.181$ | $11.04 \pm 0.117$   |

of high-level concepts respect semantic structure in a non-Euclidean geometry. The current work validates whether the resulting notions of similarity and relevance in this non-Euclidean geometry are useful for realistic downstream tasks involving the final layer embeddings and unembeddings, thereby providing empirical evidence supporting the theory. In doing so, we also depart from the prior works' focus on linear representations of hand-crafted concepts (based on word pairs and hierarchies), and we demonstrate the utility of *learned* linear representations under the non-Euclidean geometry.

**Learning Sparse and Interpretable Linear Representations**    Learning sparse and overcomplete dictionaries on word embedding spaces has a long history (Faruqui et al., 2015; Arora et al., 2018; Subramanian et al., 2018). In recent years, there has been a surge of interest in "automating and scaling" interpretability by training large SAEs on the internal layers of LLMs (e.g., Cunningham et al., 2023; Bricken et al., 2023; Templeton et al., 2024; Lieberum et al., 2024; Paulo et al., 2024; Gao et al., 2025). Despite the interest on finding high-level features in the intermediate layers, it was not known whether we could successfully learn interpretable features in the final layer of LLMs, in part because (as we argued above) using the Euclidean distance for reconstruction error is unsuitable for finding features that respect semantic structure. As for using a different objective function, we note that Braun et al. (2024) directly minimizes the KL divergence in output distributions to find features that better explain the model's behavior. In this context, we may view the approach in Appendix S1.2 as a principled approach to SAE training that is particularly suitable for LLM unembedding spaces.

**Softmax Approximation**    There exist various methods to approximate softmax to reduce the costly embedding–unembedding matrix multiplication. On the engineering side, one could shard matrices across multiple GPUs (Sutskever et al., 2014; Joulin et al., 2017). When computational resources are constrained, one can try to reduce the vocabulary size (Chen et al., 2015). For example, Morin & Bengio (2005) use a hierarchical decomposition of the vocabulary using the WordNet (Miller, 1995), while Mnih & Teh (2012); Blanc & Rendle (2018); Rawat et al. (2019) use sampling-based methods. There are also works approximating softmax gradient to speed up training (Bengio & Senécal, 2003). In this paper, we propose an alternative leveraging the softmax geometry. Another approach to approximate softmax is to reduce the embedding dimension $d$ by adapting methods for best-arm identification in multi-armed bandits (Baharav et al., 2024). A related subproblem is Maximum Inner Product Search (MIPS) (Jégou et al., 2011; Lorenzen & Pham, 2020), which can be solved using an LSH-based method. LSH-based algorithms have also been used to identify dominant entries in the attention matrices (Zandieh et al., 2023; Han et al., 2023). These approaches are largely orthogonal to this paper's approach to leverage a semantically meaningful notion of similarity.

## 7    CONCLUSIONS

The results in this paper show the benefit of studying the softmax geometry. In particular, training sparse autoencoders with a causal inner product yields more interpretable features, and leveraging a dual map between embeddings and unembeddings enables SAE-based hashing to efficiently retrieve the top-K next tokens. Although experiments are limited to Gemma models where the vocabulary size is relatively small, the empirical results do show promises that these type of approaches could offer significant computational benefits in large-vocabulary settings as the approximation only utilize a tiny fraction of the vocabulary. A separate and orthogonal direction is to apply softmax geometry to attention mechanisms and the KV cache (e.g., Kim et al., 2024).

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

# SUPPLEMENTARY MATERIALS

## S1 ADDITIONAL EXPERIMENTAL RESULTS AND DETAILS ON GEOMETRY-AWARE SPARSE AUTOENCODERS FOR UNEMBEDDINGS

### S1.1 SAE TRAINING DETAILS

Following Gao et al. (2025), we use a TopK activation function for the encoder, which sets all but the top $K$ activations to zero. (In particular, we use $k = 5$.) We extract the unembedding matrix from Gemma-2 2B (Team et al., 2024b) and train two SAEs: one using the Euclidean distance as the loss function, and the other using the norm induced by the causal inner product defined in equation 2.2 as the loss function. Note that the latter is equivalent to first applying the whitening transformation to the unembedding matrix, and then training the SAE using Euclidean distance, which is what we do in practice to avoid computing the inverse covariance matrix at each training step. Using the Adam optimizer (Kingma & Ba, 2017) with default parameters and a learning rate of $3 \times 10^{-4}$, we train batches of 8192 examples, learning for 100 epochs. The dictionary size is 4096, while the unembedding dimension is 2304. Training can be completed with one GPU. We provide illustrative examples in the following section. In addition, we evaluated Gemma-7B and Gemma-2 9B, with win-rate results reported in Figure 1.

### S1.2 ADDITIONAL RESULTS

Figures S1 to S9 show active SAE features for different tokens. The SAE trained with the causal inner product consistently produces features relevant to the target token, whereas the vanilla SAE can yield nonsensical ones. Interestingly, the vanilla SAE performs well on location-related tokens like "Chicago" or "London."

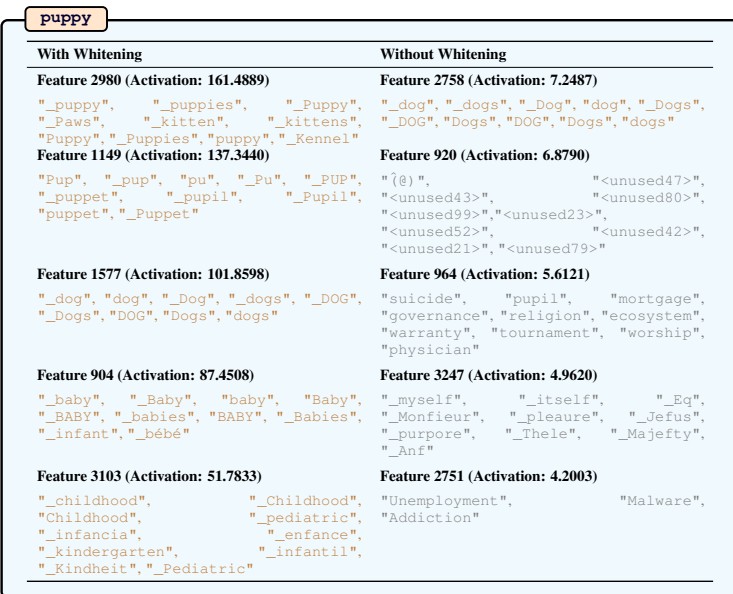

**Figure S1:** The causal inner product is necessary for the model to consistently learn meaningful features. This figure shows that with causal inner product (whitening), the top features are more semantically meaningful.

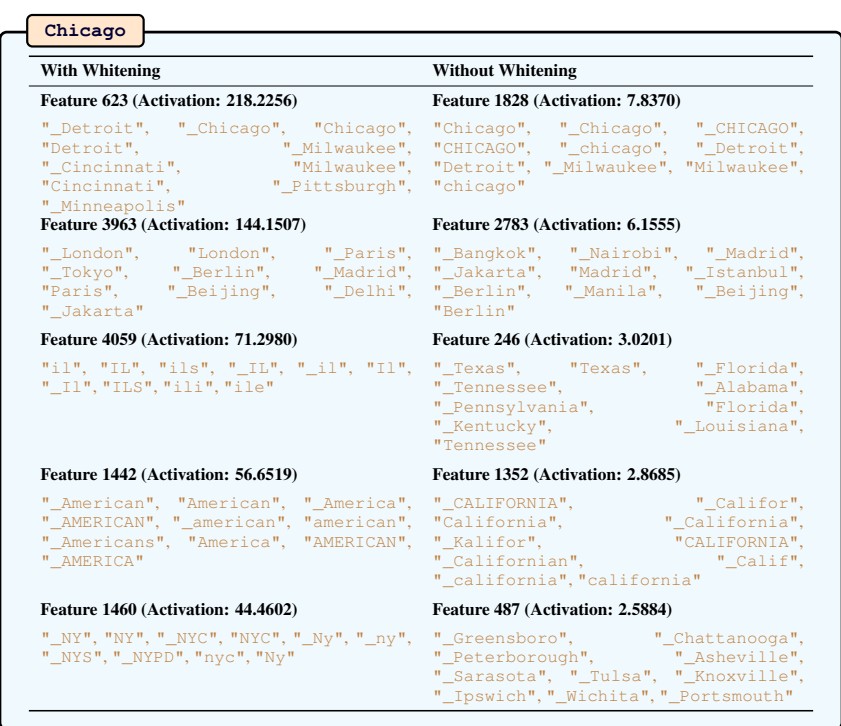

**Figure S2:** The causal inner product is necessary for the model to consistently learn meaningful features. The figure shows top active features for token "Queen".

**Figure S3:** In some cases, the Euclidean geometry seems sufficient. This is especially true for the analysis of places – such as "Chicago" and "London". The figure shows top active features for token "Chicago".

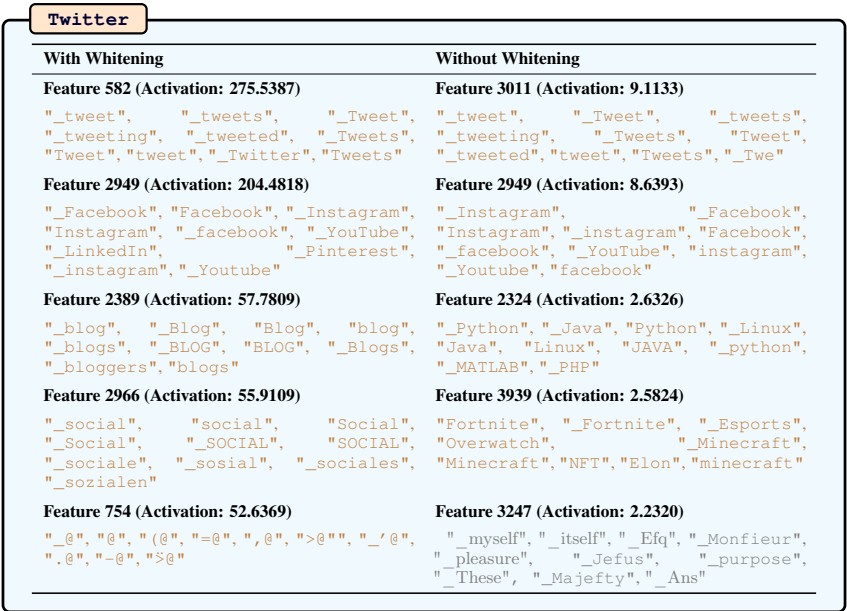

**London**

| With Whitening | Without Whitening |
|---|---|
| **Feature 3963 (Activation: 227.1938)** | **Feature 3406 (Activation: 8.6483)** |
| "_London", "London", "_Paris", "_Tokyo", "_Berlin", "_Madrid", "Paris", "_Beijing", "_Delhi", "_Jakarta" | "London", "_London", "_LONDON", "_london", "LONDON", "london", "_Londres", "_Lond", "_Лон", "_paris" |
| **Feature 2096 (Activation: 189.4191)** | **Feature 2666 (Activation: 7.0375)** |
| "_British", "British", "_Britain", "_UK", "_BRITISH", "Britain", "_british", "UK", "_brit", "british" | "_British", "British", "_Britain", "Britain", "_UK", "_BRITISH", "_british", "_brit", "BRIT", "UK" |
| **Feature 1009 (Activation: 122.3449)** | **Feature 2783 (Activation: 6.8139)** |
| "_Lon", "Lon", "LON", "_LON", "lon", "_Longitudinal", "_lon", "_longitudinal", "_Longitud", "_Lone" | "_Bangkok", "_Nairobi", "_Madrid", "_Jakarta", "Madrid", "_Istanbul", "_Berlin", "_Manila", "_Beijing", "Berlin" |
| **Feature 3464 (Activation: 85.4231)** | **Feature 2289 (Activation: 2.7443)** |
| "_Nottingham", "_Leicester", "_Bristol", "_Manchester", "_Essex", "_Southampton", "_Sheffield", "_Coventry", "_Leeds", "Bristol" | "Roberts", "Gordon", "Robertson", "Allen", "Howard", "Meyer", "Kelly", "Leslie" |
| **Feature 1008 (Activation: 49.0102)** | **Feature 1943 (Activation: 2.2330)** |
| "_Italy", "_Spain", "_France", "_Germany", "Italy", "_India", "France", "Germany", "Spain", "_Poland" | "_Spain", "_Poland", "Spain", "_Hungary", "_Brazil", "Poland", "_España", "_Portugal", "_Romania", "_Austria" |

**Figure S4:** In some cases, the Euclidean geometry seems sufficient. This is especially true for the analysis of places – such as "Chicago" and "London". The figure shows top active features for token "London".

**Twitter**

| With Whitening | Without Whitening |
|---|---|
| **Feature 582 (Activation: 275.5387)** | **Feature 3011 (Activation: 9.1133)** |
| "_tweet", "_tweets", "_Tweet", "_tweeting", "_tweeted", "_Tweets", "Tweet", "tweet", "_Twitter", "Tweets" | "_tweet", "_Tweet", "_tweets", "_tweeting", "_Tweets", "Tweet", "_tweeted", "tweet", "Tweets", "_Twe" |
| **Feature 2949 (Activation: 204.4818)** | **Feature 2949 (Activation: 8.6393)** |
| "_Facebook", "Facebook", "_Instagram", "Instagram", "_facebook", "_YouTube", "_LinkedIn", "_Pinterest", "_instagram", "_Youtube" | "_Instagram", "_Facebook", "Instagram", "_instagram", "Facebook", "_facebook", "_YouTube", "instagram", "_Youtube", "facebook" |
| **Feature 2389 (Activation: 57.7809)** | **Feature 2324 (Activation: 2.6326)** |
| "_blog", "_Blog", "Blog", "blog", "_blogs", "_BLOG", "BLOG", "_Blogs", "_bloggers", "blogs" | "_Python", "_Java", "Python", "_Linux", "Java", "Linux", "JAVA", "_python", "_MATLAB", "_PHP" |
| **Feature 2966 (Activation: 55.9109)** | **Feature 3939 (Activation: 2.5824)** |
| "_social", "social", "Social", "_Social", "_SOCIAL", "SOCIAL", "_sociale", "_sosial", "_sociales", "_sozialen" | "Fortnite", "_Fortnite", "_Esports", "Overwatch", "_Minecraft", "Minecraft", "NFT", "Elon", "minecraft" |
| **Feature 754 (Activation: 52.6369)** | **Feature 3247 (Activation: 2.2320)** |
| "_@", "@", "(@", "=@", ",@", ">@", "_'@", ".@", "-@", "ſ@" | "_myself", "_itself", "_Efq", "_Monfieur", "_pleasure", "_Jefus", "_purpose", "_These", "_Majefty", "_Ans" |

**Figure S5:** The causal inner product is necessary for the model to consistently learn meaningful features. The figure shows top active features for token "Twitter". Character U+017F is replaced with "s" for feature 3247.

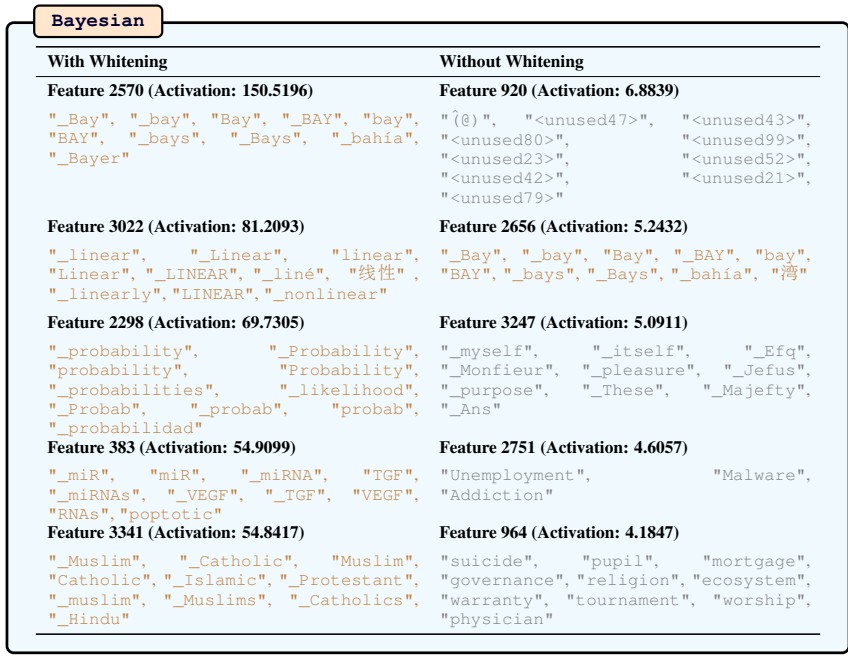

**Figure S6:** In some cases the Euclidean geometry is sufficient for semantic coherence. The figure shows top active features for token "python". The Chinese character in feature 3710 means "snake".

**Figure S7:** The causal inner product is necessary for the model to consistently learn meaningful features. The figure shows top active features for token "Bayesian". Character U+017F is replaced with "s" for feature 3247. Note that Bayes was a priest, so feature 3341 makes sense. And the Chinese characters in feature 3022 means "linear" and in 2656 the Chinese character means "Bay".

## S1.3 GPT-4O AS A JUDGE

Figure S10 shows the GPT prompt to judge SAE features.

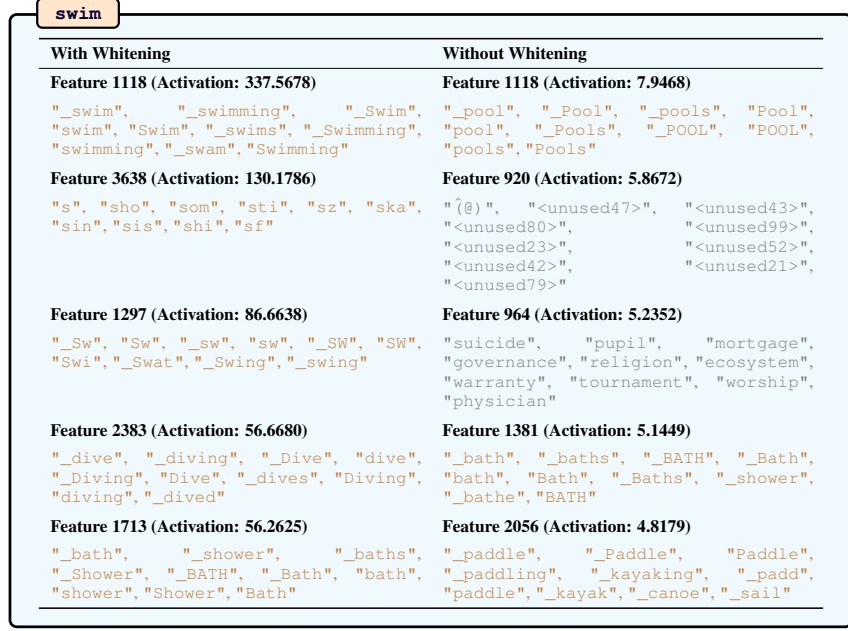

**Figure S8:** The causal inner product is necessary for the model to consistently learn meaningful features. The figure shows top active features for token "SUV". Character U+017F is replaced with "s" for feature 3247.

**Figure S9:** The causal inner product is necessary for the model to consistently learn meaningful features. The figure shows top active features for token "swim".

---

**System Prompt for GPT-4o as A Judge (SAE comparison)**

Please act as an impartial judge and evaluate which set of features best captures the meaning of the target word. Your task is to compare two explanations (A and B) and decide which one is better. Give a short verdict in the given format and provide a short explanation for your decision. Focus only on how well the explanations capture the meaning of the target word. Do not let the order or length of the explanations bias your judgment.

After providing your explanation, you must output your final verdict by strictly following this format: '[[A]]' if explanation A is substantially better, '[[B]]' if explanation B is substantially better, and '[[C]]' when there is a tie.

[Target word] {word}
[Explanation A] {response_a} [End of Explanation A]
[Explanation B] {response_b} [End of Explanation B]

---

**Figure S10:** System prompt for LLM as a Judge of Feature Explanations

## S2 Additional Experimental Results and Details on Geometry-aware Next Token Prediction

**Additional LSH results** Figures S11 to S14 show the performance on LSH based method by varying the number of hash tables and the number of hash functions per table. Overall, increasing the number of hash tables has a greater impact on performance than adding more hash functions. In particular, Figure S14 shows the results of applying LSH to whitened unembedding vectors and to the whitened exact dual representations of embeddings.

**Top-K recall** Table S1 and Table S2 show additional average top-K recall rate for Gemma-2 2B and 9B models. Because these models often assign conditional next-token probabilities near one, using the exact dual map—computed as the probability-weighted average of token unembeddings—boosts LSH's top-1 retrieval. Nonetheless, SAE with an approximately whitened dual map still outperforms LSH at recovering the remaining top tokens.

**Additional Experiments on AG News Dataset** To address potential sampling bias in the Book-Corpus dataset, we also evaluate our methods on samples from the AG News dataset (Zhang et al., 2015), using the same model and hyperparameters as before. Figure S15 shows similar trends, with the SAE method outperforming LSH-based approaches.

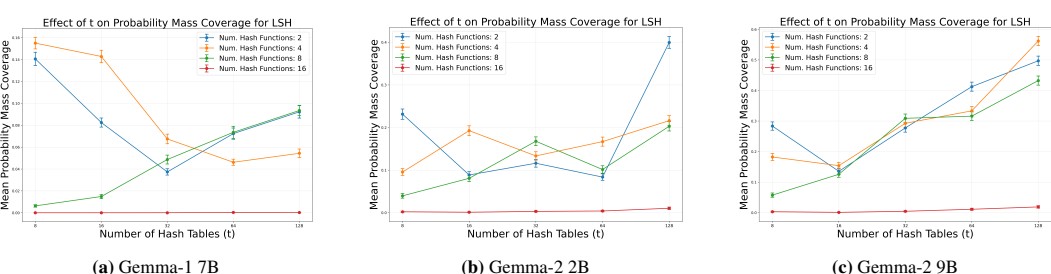

**(a)** Gemma-1 7B     **(b)** Gemma-2 2B     **(c)** Gemma-2 9B

**Figure S11:** Effect of the number of hash tables on mean probability mass coverage for LSH with varying numbers of hash functions per table. Curves correspond to 2, 4, 8 and 16 hash functions; error bars indicate standard error over 1000 samples. Panels (a), (b) and (c) show results for the Gemma-1 7B, Gemma-2 2B and Gemma-2 9B models, respectively.

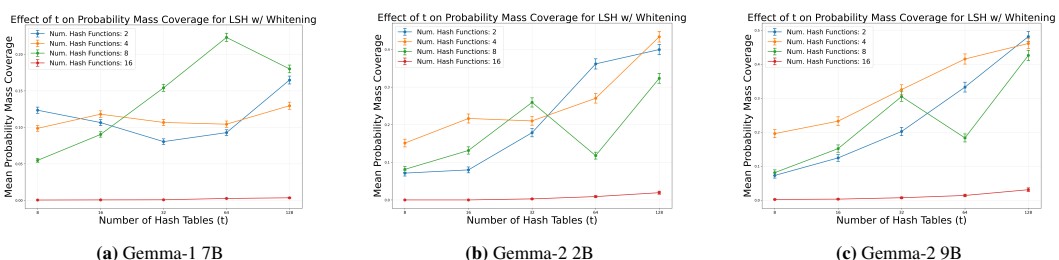

**(a)** Gemma-1 7B     **(b)** Gemma-2 2B     **(c)** Gemma-2 9B

**Figure S12:** Effect of the number of hash tables on mean probability mass coverage for whitened LSH on varying numbers of hash functions per table. Curves correspond to 2, 4, 8 and 16 hash functions; error bars indicate standard error over 1000 samples. Panels (a), (b) and (c) show results for the Gemma-1 7B, Gemma-2 2B and Gemma-2 9B models, respectively.

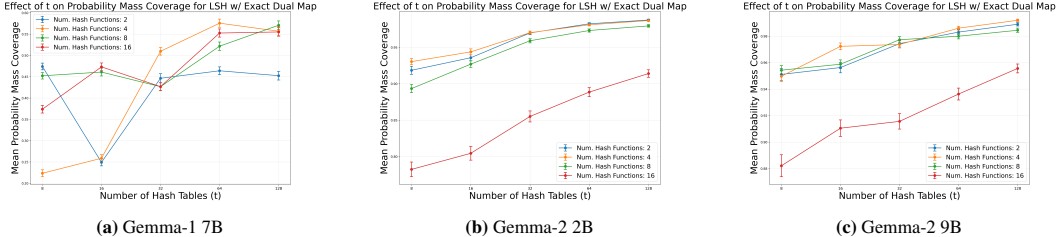

**(a)** Gemma-1 7B     **(b)** Gemma-2 2B     **(c)** Gemma-2 9B

**Figure S13:** Effect of the number of hash tables on mean probability mass coverage for LSH with exact dual map on varying numbers of hash functions per table. Curves correspond to 2, 4, 8 and 16 hash functions; error bars indicate standard error over 1000 samples. Panels (a), (b) and (c) show results for the Gemma-1 7B, Gemma-2 2B and Gemma-2 9B models, respectively.

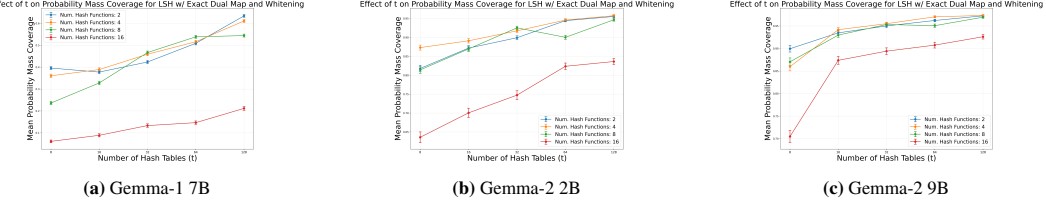

**(a)** Gemma-1 7B     **(b)** Gemma-2 2B     **(c)** Gemma-2 9B

**Figure S14:** Effect of the number of hash tables on mean probability mass coverage for LSH with exact whitened dual map on varying numbers of hash functions per table. Curves correspond to 2, 4, 8 and 16 hash functions; error bars indicate standard error over 1000 samples. Panels (a), (b) and (c) show results for the Gemma-1 7B, Gemma-2 2B and Gemma-2 9B models, respectively.

**Table S1:** The SAE-based approach outperforms LSH-based methods in retrieving the top-K tokens. The table shows the average recall rates for these top-K tokens, with standard errors computed over 1,000 samples using Gemma-2 2B model.

| Top-K | SAE w/ Approx. Dual | LSH w/ Exact Dual | LSH w/ Approx. Dual |
|---|---|---|---|
| 1 | $0.927 \pm 0.008$ | $0.999 \pm 0.001$ | $0.887 \pm 0.010$ |
| 5 | $4.345 \pm 0.029$ | $4.309 \pm 0.033$ | $3.510 \pm 0.038$ |
| 10 | $8.275 \pm 0.052$ | $8.045 \pm 0.071$ | $6.030 \pm 0.063$ |
| 20 | $15.62 \pm 0.105$ | $15.00 \pm 0.149$ | $10.08 \pm 0.106$ |

**Table S2:** The SAE-based approach outperforms LSH-based methods in retrieving the top-K tokens. The table shows the average recall rates for these top-K tokens, with standard errors computed over 1,000 samples using Gemma-2 9B model.

| Top-K | SAE w/ Approx. Dual | LSH w/ Exact Dual | LSH w/ Approx. Dual |
|---|---|---|---|
| 1 | $0.932 \pm 0.008$ | $0.993 \pm 0.003$ | $0.885 \pm 0.010$ |
| 5 | $4.209 \pm 0.032$ | $4.051 \pm 0.035$ | $3.304 \pm 0.038$ |
| 10 | $7.810 \pm 0.060$ | $7.510 \pm 0.072$ | $5.394 \pm 0.069$ |
| 20 | $14.20 \pm 0.118$ | $13.66 \pm 0.143$ | $8.679 \pm 0.117$ |

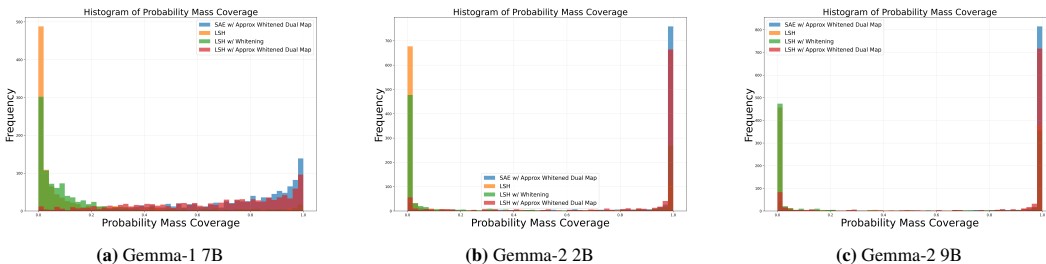

(a) Gemma-1 7B      (b) Gemma-2 2B      (c) Gemma-2 9B

**Figure S15:** The SAE-based approach consistently outperforms LSH-based methods. The figures depict histograms of probability mass convergence over 1000 samples from the AG News dataset.

## S2.1 TEXT GENERATION EXAMPLES

The prompt asking GPT-4o to generate complete sentences is shown in Figure S16. And this section shows a collection of texts generated by both default decoding and SAE-based decoding.

> **Prompt for Complete Sentences Generation**
>
> Give me 100 sentences covering very diverse topics including history, science, sports, arts, literature, entertainment, politics, economics, celebrity, etc. Each sentence has a length longer than 10 words and smaller than 200 words.

**Figure S16:** GPT-4o prompt for complete sentences generation

1134
1135
1136
1137
1138
1139
1140
1141
1142
1143
1144
1145
1146
1147
1148
1149
1150
1151
1152
1153
1154
1155
1156
1157
1158
1159
1160
1161
1162
1163
1164
1165
1166
1167
1168
1169
1170
1171
1172
1173
1174
1175
1176
1177
1178
1179
1180
1181
1182
1183
1184
1185
1186
1187

**Table S3:** Text generation by default and SAE-based approach with greedy decoding. The maximum number of new tokens is 100.

**Prompt:** The 2008 global financial crisis originated in the collapse of

| | | |
|---|---|---|
| **Gemma-1-7b** | *Default* | The 2008 global financial crisis originated in the collapse of the US housing market. The crisis was triggered by the collapse of the US housing market, which was caused by a combination of factors, including the subprime mortgage crisis, the collapse of the housing bubble, and the failure of the US government to regulate the financial sector. The crisis had a profound impact on the global economy, leading to a deep recession and a sharp decline in global trade. 

 The crisis began in the US housing market in 2007, when the subprime mortgage crisis |
| | *SAE-based* | The 2008 global financial crisis originated in the collapse of the US subprime mortgage market. The crisis was caused by a combination of factors, including the use of complex financial instruments, the lack of regulation in the financial sector, and the over-extension of credit. The crisis had a significant impact on the global economy, leading to a deep global depression and a significant loss of jobs and economic output. 

 The subprime market was a market for mortgages that were given to borrowers with poor credit histories. These borrowers were more at risk of not being able to |
| **Gemma-2-2b** | *Default* | The 2008 global financial crisis originated in the collapse of the US housing market. The crisis was triggered by the subprime mortgage crisis, which was caused by the over-issuance of subprime mortgages. The subprime mortgage crisis was caused by the over-issuance of subprime mortgages. The subprime mortgage crisis was caused by the over-issuance of subprime mortgages. The subprime mortgage crisis was caused by the over-issuance of subprime |
| | *SAE-based* | The 2008 global financial crisis originated in the collapse of the US housing market. The crisis was triggered by the subprime mortgage crisis, which was caused by the over-reliance on subprime mortgages. The subprime mortgage crisis was caused by the over-reliance on subprime mortgages. The subprime mortgage crisis was caused by the over-reliance on subprime mortgages. The subprime mortgage crisis was caused by the over-reliance on subprime loans. The subprime mortgage crisis was caused by the over-reliance on subprime loans. The subprime |
| **Gemma-2-9b** | *Default* | The 2008 global financial crisis originated in the collapse of the US housing market. The crisis was caused by a combination of factors, including the subprime mortgage crisis, the collapse of the investment bank Lehman Brothers, and the failure of the US government to regulate the financial sector. The crisis had a devastating impact on the global economy, leading to a recession in many countries and a sharp decline in stock markets. 

 The 2008 global financial crisis was caused by a combination of factors, including the subprime mortgage crisis, the collapse of the investment |
| | *SAE-based* | The 2008 global financial crisis originated in the collapse of the US subprime mortgage market. The crisis was caused by a combination of factors, including the use of complex financial instruments, the failure of financial institutions to manage risk, and the lack of regulation. The crisis had a significant impact on the global economy, with many countries experiencing a sharp decline in economic activity. The crisis also led to a number of changes in the financial system, including the introduction of new regulations and the creation of new institutions to oversee the financial sector. 

 The global financial crisis of |

**Table S4:** Text generation by default and SAE-based method with top-k and top-p sampling at a temperature 1.0. The maximum number of new tokens is 100.

**Prompt:** The 2008 global financial crisis originated in the collapse of

| | | |
|---|---|---|
| **Gemma-1-7b** | *Default* | The 2008 global financial crisis originated in the collapse of the U.S. mortgage-backed securities market. Over the past decade, the mortgage and mortgage-backed securities markets in the U.S. have recovered significantly. Despite these improvements, the U.S. housing market has not yet recovered.

The U.S. housing market has not yet fully recovered from the 2008 global financial crisis. One of the biggest problems is that home prices have not yet recovered to pre-crisis levels. This is partly because mortgage lending standards |
| | *SAE-based* | The 2008 global financial crisis originated in the collapse of the US subprime mortgage market, but the ultimate costs were much broader, with adverse consequences for the whole global economy. It has often been said that this collapse was a classic case of 'too big to fail', in which the scale of the financial institutions made them too large to be allowed to go under.

In its recent report, the Institute for Public Policy Research (IPPR) considers whether the same problem could be a danger for the UK. It states that this problem can be seen across |
| **Gemma-2-2b** | *Default* | The 2008 global financial crisis originated in the collapse of the US subprime mortgage market. The crisis was followed by a global recession, which was the worst since the Great Depression of the 1930s. The crisis was caused by a number of factors, including the subprime mortgage crisis, the collapse of the housing market, and the credit crunch. The subprime mortgage crisis was caused by the rapid growth of the subprime mortgage market in the United States. The collapse of the housing market was caused by the subprime mortgage crisis, which |
| | *SAE-based* | The 2008 global financial crisis originated in the collapse of the U.S. housing market. The crisis was caused by the over-issuance of mortgage-backed bonds, which were then used to finance the purchase of houses. The crisis was made worse by the fact that the U.S. government did not intervene to stop the crisis from spreading. The crisis was made worse by the fact that the U.S. government did not intervene to stop the crisis from spreading.

The 2008 financial crisis was caused by the collapse of |
| **Gemma-2-9b** | *Default* | The 2008 global financial crisis originated in the collapse of the US subprime mortgage market. The crisis spread to the rest of the world, and the global economy was plunged into a recession. The crisis was caused by a combination of factors, including lax lending standards, poor risk management, and a lack of regulation. The crisis had a profound impact on the global economy, and it is still being felt today.

The global financial crisis of 2008 was caused by a number of factors, including the housing bubble, the subprime mortgage crisis |
| | *SAE-based* | The 2008 global financial crisis originated in the collapse of the US subprime mortgage market. The crisis was caused by a combination of factors, including the failure of financial institutions to assess and manage risk, the growth of complex and hard-to-price financial products, and the lack of regulation and supervision of the financial system. The crisis had a significant impact on the global economy, with many countries experiencing economic slowdowns and high levels of debt. The crisis also led to a number of changes in the financial system, including increased regulation and the creation of new |

1242
1243
1244
1245
1246
1247
1248
1249
1250
1251
1252
1253
1254
1255
1256
1257
1258
1259
1260
1261
1262
1263
1264
1265
1266
1267
1268
1269
1270
1271
1272
1273
1274
1275
1276
1277
1278
1279
1280
1281
1282
1283
1284
1285
1286
1287
1288
1289
1290
1291
1292
1293
1294
1295

**Table S5:** Text generation by default and SAE-based method with top-k and top-p sampling at a temperature 0.7. The maximum number of SAE-based tokens is 100.

**Prompt:** The 2008 global financial crisis originated in the collapse of

| | | |
|---|---|---|
| **Gemma-1-7b** | *Default* | The 2008 global financial crisis originated in the collapse of the subprime mortgage market in the United States. The crisis has since spread to the rest of the world. It has triggered a global recession that has affected both developed and developing countries alike. The global financial crisis has been described as the worst economic crisis since the Great Depression of the 1930s. The crisis has had a profound impact on the global economy. It has led to a sharp decline in economic growth, an increase in unemployment, and a rise in poverty. The crisis has |
| | *SAE-based* | The 2008 global financial crisis originated in the collapse of the US subprime mortgage market. It was a direct result of the excessive risk-taking by financial institutions that was fuelled by the availability of cheap credit and the absence of effective regulation. 

 The crisis was a wake-up call for financial regulators, who were not fully up to the task of ensuring the stability of the financial system. 

 The crisis also showed the limitations of the traditional approach to financial regulation, which was based on a model of financial institutions as independent entities. 

 This model did not |
| **Gemma-2-2b** | *Default* | The 2008 global financial crisis originated in the collapse of the US subprime mortgage market. The crisis was followed by a global recession, which was the worst since the Great Depression of the 1930s. The crisis was caused by a number of factors, including the over-leveraging of the financial sector, the lack of regulation of the financial sector, and the lack of transparency in the financial sector. The crisis also led to a number of other problems, including the collapse of the housing market, the collapse of the auto industry, and |
| | *SAE-based* | The 2008 global financial crisis originated in the collapse of the US housing market. The crisis was triggered 
 by the subprime mortgage crisis, which was caused by the over-reliance on the use of credit to finance the purchase of homes. The crisis spread to other financial markets, including the stock market, and had a significant impact on the global economy. The crisis was eventually resolved through a combination of government intervention and private sector action. 

 The 2008 financial crisis was a global financial crisis that began in the United States in 200 |
| **Gemma-2-9b** | *Default* | The 2008 global financial crisis originated in the collapse of the US subprime mortgage market. The crisis was triggered by the collapse of the US subprime mortgage market, which was caused by the bursting of the US housing bubble. The crisis spread to other countries through the global financial system, causing a global economic recession. The crisis had a significant impact on the global economy, causing a decrease in economic growth, an increase in unemployment, and a decrease in stock market prices. The crisis also led to a decrease in consumer confidence and a decrease in investment. |
| | *SAE-based* | The 2008 global financial crisis originated in the collapse of the US subprime mortgage market. The crisis was caused by a combination of factors, including the rapid growth of the US economy, the low interest rates, and the high level of debt. The crisis had a significant impact on the global economy, and it led to a period of economic stagnation. 

 The 2008 global financial crisis was a major economic event that had a significant impact on the global economy. The crisis was caused by a combination of factors, including the rapid growth of the |

1296
1297
1298
1299
1300
1301
1302
1303
1304
1305
1306
1307
1308
1309
1310
1311
1312
1313
1314
1315
1316
1317
1318
1319
1320
1321
1322
1323
1324
1325
1326
1327
1328
1329
1330
1331
1332
1333
1334
1335
1336
1337
1338
1339
1340
1341
1342
1343
1344
1345
1346
1347
1348
1349

**Table S6:** Text generation by default and SAE-based method with greedy decoding. The maximum number of new tokens is 100.

**Prompt:** The U. S. Civil Rights Movement of the 1950s and 60s

| | | |
|---|---|---|
| **Gemma-1-7b** | *Default* | The U. S. Civil Rights Movement of the 1950s and 60s was a time of great change in the United States. The movement was a response to the discrimination and segregation that African Americans faced in the South. The movement was led by a number of civil rights leaders, including Martin Luther King, Jr., Rosa Parks, and John Lewis. The movement was successful in achieving a number of important goals, including the passage of the Civil Rights Act of 1964 and the Voting Rights Act of 1965.

The Civil Rights |
| | *SAE-based* | The U. S. Civil Rights Movement of the 1950s and 60s was a time of great change in the United States. The movement was a response to the discrimination and inequality that African Americans had faced for centuries. The movement was led by a number of different organizations and individuals, including the National Association for the Advancement of Colored People (NAACP), the Southern Christian Leadership Conference (SCLC), and the Student Non-DisCriminatory Action Committee (SNCC). The movement was also supported by a number of different individuals, including Martin Luther King, Jr., Malcolm |
| **Gemma-2-2b** | *Default* | The U. S. Civil Rights Movement of the 1950s and 60s was a time of great change and progress for African Americans. The movement was led by a group of activists who fought for equal rights and justice for all people. The movement was successful in many ways, but it also had its share of setbacks. One of the most important setbacks was the assassination of Martin Luther King Jr. in 1968. This assassination had a major impact on the movement, and it led to a decline in support for the movement. The assassination also led to a decline |
| | *SAE-based* | The U. S. Civil Rights Movement of the 1950s and 60s was a time of great change and progress for African Americans. The movement was led by a group of activists who were determined to end the systemic discrimination and segregation that had been in place for generations. The movement was successful in many ways, but it also had its fair share of challenges. One of the biggest challenges was the violence that was often used against the movement. The violence was often directed at the leaders of the movement, but it also included attacks on the homes and businesses of the movement's |
| **Gemma-2-9b** | *Default* | The U. S. Civil Rights Movement of the 1950s and 60s was a time of great change and progress for African Americans. It was a time when African Americans began to fight for their rights and demand equality. The movement was led by a number of prominent African American leaders, including Martin Luther King Jr., Rosa Parks, and Malcolm X. The movement was marked by a number of important events, including the Montgomery Bus Boycott, the March on Washington, and the Civil Rights Act of 1964. The movement was successful in achieving many of its goals |
| | *SAE-based* | The U. S. Civil Rights Movement of the 1950s and 60s was a time of great change and progress for African Americans. It was a time when African Americans began to fight for their rights and demand equal treatment under the law. The movement was led by a number of prominent African American leaders, including the late civil rights leader, Dr. King. The movement was successful in achieving many of its goals, including the passage of the Civil Rights Act of 1964 and the Voting Rights Act of 1965.

The Civil Rights Movement was |

**Table S7:** Text generation by default and SAE-based method with top-k and top-p sampling at a temperature 1.0. The maximum number of new tokens is 100.

**Prompt:** The U. S. Civil Rights Movement of the 1950s and 60s

| | | |
|---|---|---|
| **Gemma-1-7b** | *Default* | The U. S. Civil Rights Movement of the 1950s and 60s is one of the most important movements in the history of the United States. But how well do you know it? In this post, we offer ten key moments in the history of the movement. We also provide links to important films, documentaries, and documentaries that are available for streaming or rental on Netflix and other video streaming sites. We also offer links to sites where you can purchase the films on DVD. The films are grouped according to the eras of the Civil Rights Movement.

<h4>Key Moments |
| | *SAE-based* | The U. S. Civil Rights Movement of the 1950s and 60s was one of the most powerful movements for change in the history of the United States. From the peaceful protest of the Freedom Rides to the powerful words of Martin Luther King, Jr. to the strength of the Black Power movement, this event helped to shape the course of American history.

We celebrate this powerful movement and its leaders by displaying a variety of photographs, posters and objects. |
| **Gemma-2-2b** | *Default* | The U. S. Civil Rights Movement of the 1950s and 60s was a time of great change in the United States. The movement was a response to the injustices that African Americans faced in the South. The movement was led by Martin Luther King Jr. and other civil rights activists. The movement was successful in getting the Civil Rights Act of 1964 passed. The act banned discrimination in public places and employment. The movement also led to the passage of the Voting Rights Act of 1965. The act made it easier for African Americans to vote |
| | *SAE-based* | The U. S. Civil Rights Movement of the 1950s and 60s was a time of great change and progress for African Americans. The movement was led by a number of important figures, including Martin Luther King Jr., Malcolm X, and Rosa Parks. The movement was successful in its goal of achieving civil rights for African Americans, and it helped to pave the way for the passage of the Civil Rights Act of 1964.

The Civil Rights Movement was a time of great change and progress for African Americans. The movement was led by a number of important figures |
| **Gemma-2-9b** | *Default* | The U. S. Civil Rights Movement of the 1950s and 60s was a time of great change and progress for African Americans. It was a time when African Americans began to fight for their rights and equality. The Civil Rights Act of 1964 was a major piece of legislation that helped to end discrimination against African Americans. The act made it illegal to discriminate against people based on their race, color, religion, sex, or national origin. The act also helped to create equal opportunity for African Americans in education, employment, and housing. The Civil Rights Act |
| | *SAE-based* | The U. S. Civil Rights Movement of the 1950s and 60s was a time of great change and progress for African Americans. It was a time when people of all races came together to fight for equal rights and justice. The movement was led by a number of brave and courageous individuals, including the late civil rights leader, Dr. King. The movement was successful in achieving many of its goals, including the passage of the Civil Rights Act of 1964 and the Voting Rights Act of 1965. The movement also helped to bring about an |

S13

**Table S8:** Text generation by default and SAE-based method with top-k and top-p sampling at a temperature 0.7. The maximum number of new tokens is 100.

**Prompt:** The U. S. Civil Rights Movement of the 1950s and 60s

| | | |
|---|---|---|
| **Gemma-1-7b** | *Default* | The U. S. Civil Rights Movement of the 1950s and 60s was a time of incredible courage and bravery. The sacrifices of those who fought for the equality of all citizens was extraordinary. The Civil Rights Movement was a time when a group of dedicated individuals fought for the rights of all Americans to be treated equally. The Movement was a time when many people lost their lives, their jobs, and their homes. The Civil Rights Movement was a time when the United States was a better place for all Americans.

<h2>The Civil Rights Movement</h2>

The |
| | *SAE-based* | The U. S. Civil Rights Movement of the 1950s and 60s is one of the most important and best-known social movements in American history. The movement has been the subject of countless books, articles, films, and television programs. The movement is often portrayed as a unified and non-partisan effort, with civil rights leaders working together to achieve their goals. However, the reality is that the U. S. Civil Rights Movement was not as unified as it is often portrayed. In fact, there were many different factions within the movement, each with their own goals |
| **Gemma-2-2b** | *Default* | The U. S. Civil Rights Movement of the 1950s and 60s was a time of great change and progress for African Americans. The movement was led by a group of activists who fought for equal rights and justice for all people. The movement was successful in many ways, but it also had its share of challenges. One of the biggest challenges the movement faced was the violence that was used against them. The violence was used by white people to try and stop the movement from happening. The violence was also used by the government to try and stop the movement from happening. The |
| | *SAE-based* | The U. S. Civil Rights Movement of the 1950s and 60s was a time of great change and progress for African Americans. The movement was led by a group of activists who worked to end segregation and discrimination in all areas of American life. The movement was successful in many ways, but it also had its fair share of challenges. One of the most important challenges that the movement faced was the issue of violence. There were many instances of violence against African Americans during this time, and it was often used as a way to intimidate and suppress the movement. The violence was |
| **Gemma-2-9b** | *Default* | The U. S. Civil Rights Movement of the 1950s and 60s was a time of great change and progress for African Americans. It was a time when African Americans fought for their rights and equality. The movement was led by many different people, including Martin Luther King Jr., Rosa Parks, and Malcolm X. The movement was successful in many ways, including the passage of the Civil Rights Act of 1964 and the Voting Rights Act of 1965. The movement also helped to change the way that African Americans were treated in society.

The |
| | *SAE-based* | The U. S. Civil Rights Movement of the 1950s and 60s was a time of great change and progress for African Americans. It was a time when people of all races and backgrounds came together to fight for equal rights and justice. The movement was led by a number of prominent figures, including the late civil rights leader, Dr. King. The movement was successful in achieving many of its goals, including the passage of the Civil Rights Act of 1964 and the Voting Rights Act of 1965. The movement also helped to bring about an |

1458
1459
1460
1461
1462
1463
1464
1465
1466
1467
1468
1469
1470
1471
1472
1473
1474
1475
1476
1477
1478
1479
1480
1481
1482
1483
1484
1485
1486
1487
1488
1489
1490
1491
1492
1493
1494
1495
1496
1497
1498
1499
1500
1501
1502
1503
1504
1505
1506
1507
1508
1509
1510
1511

**Table S9:** Text generation by default and SAE-based method with greedy decoding. The maximum number of new tokens is 100.

**Prompt:** The complexities of international relations often involve

| | | |
|---|---|---|
| **Gemma-1-7b** | *Default* | The complexities of international relations often involve a number of different actors, each with their own interests and agendas. In the case of the United States and China, the relationship between the two countries is often seen as a zero-sum game, with each side vying for dominance and influence. However, there are other actors who play a role in shaping the relationship between the two countries, including the European Union (EU). 

The EU is a major player in the global economy and has a significant presence in both the United States and China. The |
| | *SAE-based* | The complexities of international relations often involve a number of different actors, each with their own interests and motivations. In the case of the United States and China, the relationship between the two countries is often seen as a zero-sum game, where one country's gain is the other's loss. However, this is not always the case. In fact, there are a number of ways in which the United States and China can work together to achieve mutual benefits. One area where the two countries can cooperate is in the area of trade |
| **Gemma-2-2b** | *Default* | The complexities of international relations often involve the need to balance the interests of different countries. In the case of the United States and China, the two countries have been engaged in a complex relationship that has seen them compete, cooperate, and even clash. The United States has been China's largest trading partner for decades, and the two countries have a long history of cooperation in areas such as trade, investment, and technology. However, the two countries have also been engaged in a number of disputes, including territorial disputes in the South China Sea |
| | *SAE-based* | The complexities of international relations often involve the need to balance the interests of different countries. In the case of the United States and China, the two countries have been involved in a long-running trade war that has had a significant impact on both economies. The United States has been particularly concerned about China's trade practices, which it believes are not fair and do not follow international trade rules. China, on the other hand, has been concerned about the United States' trade policies, which it believes are protectionist and do not allow for |
| **Gemma-2-9b** | *Default* | The complexities of international relations often involve a delicate balance of power and influence. In the case of the United States and China, the relationship has been characterized by both cooperation and competition. As the world's two largest economies, their interactions have far-reaching implications for global stability and prosperity. 

One of the key areas of cooperation between the United States and China is trade. Both countries have benefited from the exchange of goods and services, with China being a major trading partner for the United States. However, trade tensions have also arisen, |
| | *SAE-based* | The complexities of international relations often involve a delicate balance of power and influence. In the case of the United States and China, the relationship has been a subject of intense scrutiny and speculation. One question that has arisen is whether the United States is currently in a position of power over China. This article delves into the various factors that contribute to the power equation between these two global superpowers, examining their economic, military, and diplomatic strengths and weaknesses. 

<h2>The Economic Powerhouse: China's Economic Growth and Impact. |

1512
1513
1514
1515
1516
1517
1518
1519
1520
1521
1522
1523
1524
1525
1526
1527
1528
1529
1530
1531
1532
1533
1534
1535
1536
1537
1538
1539
1540
1541
1542
1543
1544
1545
1546
1547
1548
1549
1550
1551
1552
1553
1554
1555
1556
1557
1558
1559
1560
1561
1562
1563
1564
1565

**Table S10:** Text generation by default and SAE-based method with top-k and top-p sampling at a temperature 1.0. The maximum number of new tokens is 100.

**Prompt:** The complexities of international relations often involve

| | | |
|---|---|---|
| **Gemma-1-7b** | *Default* | The complexities of international relations often involve more than just the two governments involved. In fact, the dynamics of international relations may be greatly influenced by non-governmental organizations, non-governmental entities, and international actors and stakeholders.

It is for this reason that the United Nations established the Non-Governmental Liaison Section (NGLS), which is a dedicated division for the purpose of facilitating partnerships between the UN and NGOs.

The NGLS is the focal point for the UN's contact with NGOs. There are more than |
| | *SAE-based* | The complexities of international relations often involve the movement of multiple parties and actors. These relationships involve states, non-governmental organisations, private companies, or individuals. Understanding how these different groups and parties work and how they interact with one another is essential for a successful business.

One important aspect of international relations is the concept of a network. A network is a group of people or organisations that interact with one another to achieve a common goal. In international relations, networks can be used to facilitate the exchange of information, resources, and support between |
| **Gemma-2-2b** | *Default* | The complexities of international relations often involve the need to balance the interests of different countries. In the case of the United States and China, the relationship between the two countries has been characterized by a complex interplay of competition, cooperation, and mutual dependence.

One of the most significant aspects of the US-China relationship is the issue of trade. The United States has long been a major trading partner with China, and the two countries have a large trade deficit. This has led to concerns about the impact of Chinese imports on the US economy and |
| | *SAE-based* | The complexities of international relations often involve the need to balance the interests of different countries. In the case of the United States and China, the relationship has been a source of tension and cooperation. The United States has been a key player in the global economy, while China has been a major player in the global market. The two countries have been involved in a number of trade disputes, with the United States imposing various sanctions on China. The United States has also been involved in the development of technology, with China being a major player in the |
| **Gemma-2-9b** | *Default* | The complexities of international relations often involve a delicate balancing act between the interests of different countries. In the case of the United States and China, the relationship has been marked by both cooperation and competition. While the two nations have collaborated on various fronts, including trade and climate change, they have also engaged in a strategic rivalry, particularly in the realm of technology and national security.

One of the key areas of contention between the U.S. and China is the issue of technology. The U.S. has expressed concerns about China' |
| | *SAE-based* | The complexities of international relations often involve a delicate balance of power and influence. In the realm of global politics, the United States and China have been at the center of a power shift that has far-reaching implications for the world order. This article delves into the dynamics of this power shift, highlighting the factors that have led to China's growing influence and the challenges it presents for the United States.

<h2>The Rise of China: A Global Powerhouse in the Making? |

**Table S11:** Text generation by default and SAE-based method with top-k and top-p sampling at a temperature 0.7. The maximum number of new tokens is 100.

**Prompt:** The complexities of international relations often involve

| | | |
|---|---|---|
| **Gemma-1-7b** | *Default* | The complexities of international relations often involve different ways of looking at the same set of facts. In the Middle East, it is often assumed that the two-state solution is the only way forward to resolve the Israeli-Palestinian conflict. Yet, it is not the only option.

While a two-state solution might be the most likely way forward, it is not the only way forward. It is not the only viable option, nor is it the only solution that is politically feasible. It is not even the only option that is morally |
| | *SAE-based* | The complexities of international relations often involve multiple players. In this case, the United States, Canada, and the United Kingdom are all involved in the dispute.

The United States and Canada are the two countries that are most directly involved in the dispute. The United States is the country that is being sued by Canada for its failure to comply with the terms of the North American Free Trade Agreement (NAFTA). The United States has also been involved in the dispute because it is one of the countries that has been accused of violating human rights in |
| **Gemma-2-2b** | *Default* | The complexities of international relations often involve the need to balance the interests of multiple countries. In the case of the United States and China, the relationship has been marked by a series of trade disputes, geopolitical tensions, and diplomatic challenges.

The United States has been concerned about China's trade practices, which have been seen as unfair and predatory. The U.S. has imposed tariffs on Chinese goods, and China has retaliated with its own tariffs. This has led to a trade war that has affected both countries' economies. |
| | *SAE-based* | The complexities of international relations often involve the need to balance the interests of different countries. In the case of the United States and China, the relationship has been a source of tension and competition for decades. However, there are also areas where the two countries can work together to achieve common goals.

One such area is the fight against climate change. Both the United States and China are major contributors to global carbon pollution, and both countries have made commitments to reduce their carbon pollution. The United States has set a goal of reducing its carbon pollution |
| **Gemma-2-9b** | *Default* | The complexities of international relations often involve a delicate balance of power, alliances, and strategic interests. In the ever-evolving landscape of global politics, the United States and China have emerged as two of the most influential nations, each with its own unique set of goals and aspirations. As these two superpowers navigate the intricate web of international relations, the question arises: Who is the more powerful nation between the United States and China?

<h2>The United States: A Global Powerhouse</h2>

The United States, often referred to as |
| | *SAE-based* | The complexities of international relations often involve a delicate balance of power and influence. In the case of the United States and China, the relationship between the two nations has been a subject of intense scrutiny and speculation. One question that has arisen is whether China has the ability to influence the United States through its economic power. This article delves into the dynamics of this relationship, exploring the extent to which China's economic influence can be wielded over the United States.

<h2>The Economic Powerhouse: China's Economic Strength
(h2 |

1620
1621
1622
1623
1624
1625
1626
1627
1628
1629
1630
1631

**Table S12:** Text generation by default and SAE-based method with greedy decoding. The maximum number of new tokens is 100.

**Prompt:** Streaming services have revolutionized the way people consume television

| | | |
|---|---|---|
| **Gemma-1-7b** | *Default* | Streaming services have revolutionized the way people consume television and movies. With the rise of services like Netflix, Hulu, and Amazon Prime Video, viewers have more options than ever before. But with so many choices, it can be difficult to decide which service is right for you. In this article, we'll compare Netflix vs Hulu vs Amazon Prime Video to help you make an informed decision.

h2>Netflix</h2>

Netflix is one of the most popular streaming services in the world. It offers a wide variety of TV shows and movies, including original content |
| | *SAE-based* | Streaming services have revolutionized the way people consume television and movies. With the rise of services like Netflix, Hulu, and Amazon Prime Video, viewers have more options than ever before. But with so many choices, it can be difficult to decide which service is right for you. In this article, we'll compare and contrast the three most popular services to help you make an informative decision.

<h2>Content Selection: What's Available on Each Service?

When it comes to content, each service has its own unique selection. For example, |
| **Gemma-2-2b** | *Default* | Streaming services have revolutionized the way people consume television. With the rise of Netflix, Hulu, and Amazon Prime Video, it's become easier than ever to watch your favorite shows and movies on-demand. But what if you want to watch your favorite shows on your computer?

There are a few different ways to do this. You can use a streaming service like Netflix or Hulu, or you can use a third-party website like Popcorn Time.

If you're looking for a way to watch your favorite shows on your computer, there |
| | *SAE-based* | Streaming services have revolutionized the way people consume television. With the rise of on-demand content, viewers can now watch their favorite shows and movies whenever they want, without having to wait for a specific time slot. This has led to a surge in the number of people who are subscribing to these services, and it has also created a new market for content creators.

One of the biggest benefits of streaming services is that they offer a wide variety of content. Whether you're looking for the latest hit show or a classic movie, you can find |
| **Gemma-2-9b** | *Default* | Streaming services have revolutionized the way people consume television. With the rise of Netflix, Hulu, and Amazon Prime, viewers can now watch their favorite shows and movies whenever and wherever they want. But what about the classic TV shows that we all grew up watching? Are they still available to stream? The answer is yes! Many classic TV shows are available to stream on a variety of platforms. You can find them on Netflix, Hulu, Amazon Prime, and even on some cable providers. So, if you're looking to relive your childhood or introduce |
| | *SAE-based* | Streaming services have revolutionized the way people consume television. With the rise of Netflix, Hulu, and Amazon Prime, viewers can now watch their favorite shows and movies whenever and wherever they want. But what about the classic TV shows that we all grew up with? Are they available on streaming services? The answer is yes! Many classic TV shows are available on streaming services, and they're a great way to relive your childhood or introduce your kids to some of the best shows ever made. Here are some of the best classic TV shows that you can |

1669
1670
1671
1672
1673

**Table S13:** Text generation by default and SAE-based method with top-k and top-p sampling at a temperature 1.0. The maximum number of new tokens is 100.

**Prompt:** Streaming services have revolutionized the way people consume television

| | | |
|---|---|---|
| **Gemma-1-7b** | *Default* | Streaming services have revolutionized the way people consume television. There are dozens of streaming services available in 2022, with some of the most popular choices being Netflix, Hulu, HBO Max, and more. These services offer subscribers the opportunity to watch a wide range of content, from popular movies and television shows to exclusive original programming. While these services have made it easier than ever to find the content you want to watch, it's important to know which ones offer the best value. In this blog post, we'll be looking at |
| | *SAE-based* | Streaming services have revolutionized the way people consume television and movies. With the ability to watch a show on-demand at any time, it's no wonder that the number of people who stream television and movies on a daily basis has increased. This article will explore the number of people who stream television and movies on a daily basis, how this number is growing, and the implications for the media industry. From the impact on viewership and audience size, to the rise of new platforms, this article will examine the data and explore how the numbers are |
| **Gemma-2-2b** | *Default* | Streaming services have revolutionized the way people consume television. With the rise of Netflix, Hulu, and Amazon Prime Video, it's easier than ever to watch your favorite shows and movies on-demand. But what if you want to watch your favorite shows on your computer?

There are a few ways to do this. One option is to use a streaming service like Netflix or Hulu. Another option is to use a VPN (virtual private network) to access content from other countries.

If you're looking for a way to watch your favorite |
| | *SAE-based* | Streaming services have revolutionized the way people consume television, and the latest addition to the list is the Disney+ platform. The platform is a one-stop destination for all things Disney, including movies, TV shows, and even original content.

One of the most popular shows on Disney+ is the "The World According to Jeff Goldbum," which is hosted by the famous actor and comedian Jeff Goldbum. The show is a travel show that takes the viewers on a journey around the world, as Goldbum explores different cultures and experiences.

The |
| **Gemma-2-9b** | *Default* | Streaming services have revolutionized the way people consume television. With the advent of Netflix, Hulu, and Amazon Prime, viewers can now watch their favorite shows and movies on demand, without having to wait for a new episode to air on traditional television. This has led to a surge in popularity for streaming services, as people are able to watch their favorite shows and movies whenever they want, without having to worry about missing an episode.

Streaming services have also led to a decrease in the number of people who subscribe to traditional cable or satellite television. With the |
| | *SAE-based* | Streaming services have revolutionized the way people consume television and movies. With the ability to watch content on-demand, viewers can now enjoy their favorite shows and films at their own pace. However, with so many options available, it can be difficult to know which service is right for you. In this article, we will compare two of the most popular streaming services: Netflix and Amazon Prime Video. We will look at the differences in their content libraries, pricing, and features to help you decide which service is best for your needs.

The Netflix and |

**Table S14:** Text generation by default and SAE-based method with top-k and top-p sampling at a temperature 0.7. The maximum number of new tokens is 100.

**Prompt:** Streaming services have revolutionized the way people consume television

| | | |
|---|---|---|
| **Gemma-1-7b** | *Default* | Streaming services have revolutionized the way people consume television. The emergence of new services has also made it difficult for consumers to decide which service is best for them. In this post, we will take a look at the pros and cons of the two most popular streaming services, Netflix and Hulu.

<h2>Netflix vs Hulu: What Is the Difference?</h2>

Netflix and Hulu are two of the most popular streaming services on the market. They both offer a wide selection of movies and TV shows, but there are some key differences between them.

Netflix is |
| | *SAE-based* | Streaming services have revolutionized the way people consume television, but what does that mean for the future of television? With the rise of on-demand services like Netflix and Hulu, traditional television networks are being forced to adapt. In this article, we'll explore how these changes are shaping the future of television and what it means for viewers.

One of the most significant changes is the rise of original content. With the success of shows like "House of Cards" and "Orange Is the New Black," networks are now producing more and more original content |
| **Gemma-2-2b** | *Default* | Streaming services have revolutionized the way people consume television and movies. With so many options available, it can be difficult to know which service is right for you. In this article, we will compare two popular streaming services: Netflix and Hulu. We will discuss the pros and cons of each service, as well as which one is right for you.

Netflix is a subscription-based streaming service that offers a wide variety of movies and TV shows. It is available in the United States, Canada, and several other countries. Netflix has a large library of |
| | *SAE-based* | Streaming services have revolutionized the way people consume television. With the rise of on-demand content, viewers can now watch their favorite shows and movies whenever they want, without having to wait for a specific time or channel. This has led to a surge in the number of people subscribing to these services, and it's no secret that the competition is getting tougher.

One of the most popular services is Netflix, which has been around for over a decade and has become a cultural phenomenon. However, with the rise of new services like Disney+, HBO |
| **Gemma-2-9b** | *Default* | Streaming services have revolutionized the way people consume television. With the rise of Netflix, Hulu, and Amazon Prime, viewers can now watch their favorite shows and movies on demand, without having to wait for them to air on traditional television networks. This has led to a shift in the way television is produced, with more shows being made specifically for streaming services.

One of the biggest advantages of streaming services is the ability to watch shows and movies on demand. This means that viewers can watch their favorite shows whenever they want, without having to wait for them |
| | *SAE-based* | Streaming services have revolutionized the way people consume television. With the rise of Netflix, Hulu, and Amazon Prime, viewers can now watch their favorite shows and movies whenever and wherever they want. But what about the classic TV shows that we grew up with? Are they still available to stream? The answer is yes! Many classic TV shows are now available to stream on a variety of platforms. From the original Star Trek to The Office, there's something for everyone. So, if you're looking to relive your childhood or discover a new favorite |

**Table S15:** Text generation by default and SAE-based method with greedy decoding. The maximum number of new tokens is 100.

**Prompt:** Climate change, driven by human activities, poses a

| | | |
|---|---|---|
| **Gemma-1-7b** | *Default* | Climate change, driven by human activities, poses a serious threat to the world's oceans. The oceans are warming, becoming more acidic, and losing oxygen. These changes are affecting marine ecosystems and the people who depend on them.

The ocean is a vast and complex system, and it is difficult to predict how it will respond to climate change. However, we do know that the ocean is already changing in response to rising temperatures and increasing levels of carbon dioxide in the atmosphere.

The ocean is warming at an unprecedented rate. This is causing the |
| | *SAE-based* | Climate change, driven by human activities, poses a serious threat to the world's natural resources and the people who depend on them. The impacts of climate change are already being felt in many parts of the world, and the situation is only expected to get worse in the coming years.

One of the most significant impacts of climate change is the loss of natural resources. As temperatures rise, many species of plants and animals are unable to adapt to the changing conditions and are at risk of extinction. This loss of diversity has far-reaching consequences for the |
| **Gemma-2-2b** | *Default* | Climate change, driven by human activities, poses a serious threat to the world's biodiversity. The impacts of climate change on biodiversity are expected to be particularly severe in the tropics, where the majority of the world's biodiversity is found. The impacts of climate change on biodiversity are expected to be particularly severe in the tropics, where the majority of the world's biodiversity is found. The impacts of climate change on biodiversity are expected to be particularly severe in the tropics, where the majority of the world's biodiversity is found. The impacts of climate |
| | *SAE-based* | Climate change, driven by human activities, poses a serious threat to the world's food security. The Intergovernmental Panel on Climate Change (IPCC) has warned that the world's food production will be affected by climate change. The report also states that the world's food production will be affected by climate change. The report also states that the world's food production will be reduced by 10% by 2050. The report also states that the world's food production will be reduced by 10% by |
| **Gemma-2-9b** | *Default* | Climate change, driven by human activities, poses a serious threat to the environment and human health. The global warming potential of greenhouse gases (GHGs) is a key indicator of the impact of GHG emissions on climate change. The global warming potential of a GHG is defined as the ratio of the radiative forcing of a unit mass of the GHG to the radiative forcing of a unit mass of carbon dioxide ($CO_2$) over a specified time horizon. The global warming potential of a GHG is determined by its radiative forcing, atmospheric lifetime, and the radiative |
| | *SAE-based* | Climate change, driven by human activities, poses a serious threat to the environment and human health. The global average temperature has increased by 1.1°C since the late 19th century, and the rate of increase has accelerated in recent decades. The effects of climate change are already being felt, with more frequent and intense extreme weather events, such as heatwaves, droughts, and floods. These events can have a significant impact on human health, particularly in terms of heat-related illnesses, water-borne diseases, and mental health. |

**Table S16:** Text generation by default and new method with top-k and top-p sampling at a temperature 1.0. The maximum number of new tokens is 100.

**Prompt:** Climate change, driven by human activities, poses a

| | | |
|---|---|---|
| **Gemma-1-7b** | *Default* | Climate change, driven by human activities, poses a severe and persistent threat to life on Earth. The recent report by the United Nations Intergovernmental Panel on Climate Change (IPCC) paints a picture of a worsening climate emergency.

The consequences of unchecked climate change would be devastating: floods, droughts, famines, loss of biodiversity, the submergence of coastal cities. In short, a collapse of civilization.

However, the report, while sounding a note of hope, has been met with skepticism by some observers who argue that the scale of the problem |
| | *New* | Climate change, driven by human activities, poses a serious threat to the global population and to the environment. The issue is of major concern in the context of a global population of 7.7 billion people and an estimated 9.7 billion by 2050. Many of these people will live in cities, and this presents a huge challenge in terms of infrastructure, resources and governance.

In this context, the role of local government is extremely important. Local governments are on the front line of the climate crisis and have a major role |
| **Gemma-2-2b** | *Default* | Climate change, driven by human activities, poses a serious threat to the world's biodiversity. The United Nations Convention on Biological Diversity (CBD) has set a target of protecting 30% of the world's land and ocean by 2030. However, the current rate of land and sea protection is far from meeting this target.

The CBD's 15th Conference of the Parties (COP15) is scheduled to take place in Kunming, China, from October 11 to 15, |
| | *New* | Climate change, driven by human activities, poses a serious threat to the health of the planet and the well-being of its inhabitants. The effects of climate change are already being felt around the world, with extreme weather events becoming more frequent and more powerful. The most recent example is the heat wave that hit Europe in July 2022, which led to the death of thousands of people.

The Intergovernmental Panel on Climate Change (IPCC) has issued a report warning that climate change is already having a major impact on human health, with |
| **Gemma-2-9b** | *Default* | Climate change, driven by human activities, poses a serious threat to the environment and human health. The global temperature has increased by 1.1°C since the pre-industrial era, and the Intergovernmental Panel on Climate Change (IPCC) predicts that the temperature will rise by 1.5°C by 2030. The IPCC also predicts that the global temperature will rise by 2°C by 2050. The increase in temperature will lead to more extreme weather events, such as heat waves, droughts, |
| | *New* | Climate change, driven by human activities, poses a serious threat to the world's food security. The effects of climate change on agriculture are already being felt in many parts of the world, and they are expected to become more severe in the future.

The most significant impacts of climate change on agriculture are likely to be felt in the form of changes in temperature and water availability. These changes will have a direct impact on crop production, as well as on the availability of water for irrigation.

In addition, climate change is also likely to increase the |

**Table S17:** Text generation by default and new method with top-k and top-p sampling at a temperature 0.7. The maximum number of new tokens is 100.

**Prompt:** Climate change, driven by human activities, poses a

| | | |
|---|---|---|
| **Gemma-1-7b** | *Default* | Climate change, driven by human activities, poses a threat to the survival of many species of animals. Scientists have now warned that the global decline in amphibian populations is a major indicator of the impending crisis.

A study published in the journal Proceedings of the Royal Society B: Biological Sciences says that more than 40% of the world's amphibian species are threatened with extinction.

"This is a warning sign for the health of the environment and all of life on Earth," said co-author of the study Dr. Frank Andreza |
| | *New* | Climate change, driven by human activities, poses a major threat to global security. In the context of this threat, the role of the military in addressing the issue is an important question.

The military is a large user of fossil fuels and has a significant environmental impact. The military also has a major role in providing security in the face of climate change.

This report examines the current and future roles of the military in addressing climate change. It focuses on the implications of climate change for military operations, the military's role in addressing climate change, |
| **Gemma-2-2b** | *Default* | Climate change, driven by human activities, poses a serious threat to the global environment. The effects of climate change are already being felt in many parts of the world, including increased frequency and intensity of extreme weather events, rising sea levels, and changes in precipitation patterns.

Climate change is a complex and multifaceted issue that requires a comprehensive and coordinated response from governments, businesses, and individuals. It is a global challenge that requires a global solution.

The effects of climate change are already being felt in many parts of the world, including increased frequency and intensity |
| | *New* | Climate change, driven by human activities, poses a serious threat to the global environment. The Intergovernmental Panel on Climate Change (IPCC) has warned that the world is on track to exceed the 1.5°C temperature rise target set by the 2015 Paris Agreement. This means that the world is likely to experience more extreme weather events, such as heatwaves, storms, and sea level rise, which will have a significant impact on human health and the environment.

The United Nations Environment Programme (UNEP) has also warned that |
| **Gemma-2-9b** | *Default* | Climate change, driven by human activities, poses a serious threat to the environment and human health. The effects of climate change are already being felt in many parts of the world, including increased temperatures, more frequent and intense extreme weather events, rising sea levels, and changes in precipitation patterns. These changes can have a significant impact on ecosystems, agriculture, water resources, and human health.

One of the most significant impacts of climate change is the increase in global temperatures. The average global temperature has risen by about 1.1 degrees Celsius since the pre |
| | *New* | Climate change, driven by human activities, poses a serious threat to the world's environment and economy. The world's average temperature has increased by 1.1°C since the late 19th century, and the rate of increase is increasing. The world's average temperature is expected to rise by 1.5°C by 2030, and by 2°C by 2050.

The effects of climate change are already being felt around the world. The frequency and intensity of extreme |

## S2.2 GPT-4O AS A JUDGE

We further use GPT-4o as a judge to compare fluency of paragraph pairs generated by default and new methods. The system prompt is displayed in Figure S17. Additional win rate comparison for sampling based method is shown in Figure S18. We also present some examples to showcase the pairwise comparison between vanilla model generation and SAE-based generation, and the LLM judgement.

---

**System Prompt for GPT-4o as A Judge**

Please act as an impartial judge and evaluate the quality of snippets of paragraph generated by two language models. Your task is to decide which paragraph sounds more fluent, coherent, and human-like. Focus only on the fluency of the paragraphs. Ignore correctness, factual accuracy, helpfulness, or topic relevance unless they directly affect naturalness. Do not focus on minor differences. Do not allow the position of the paragraphs or the length of the paragraphs to bias your judgment. Do not allow abrupt ending of the paragraph to bias your judgment because the paragraphs are only snippets, not necessarily complete sentences. You will receive a prompt (start of the sentence) and two paragraphs starting with this prompt (A and B, order randomized). Provide a short explanation for your decision focused solely on fluency, referring to specific phrases or stylistic features if helpful. After providing your explanation, you must output your final verdict by strictly following this format: '[[A]]' if paragraph A is substantially better, '[[B]]' if paragraph B is substantially better, and '[[C]]' when both paragraphs are fluent.
[Prompt] {prompt}
[Paragraph A] {paragraph_a} [End of Paragraph A]
[Paragraph B] {paragraph_b} [End of Paragraph B]

**Figure S17:** System prompt for LLM as A Judge

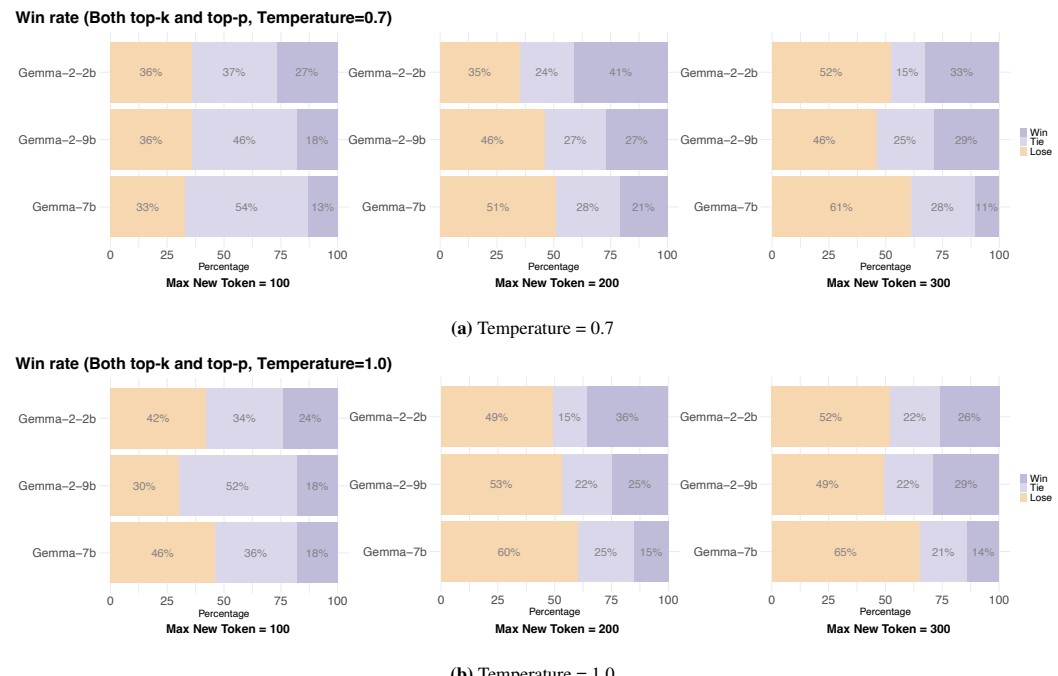

(a) Temperature = 0.7

(b) Temperature = 1.0

**Figure S18:** The SAE-based approach matches the default top-$p$/top-$k$ sampling method for sentence generation. The graphs show GPT-4o–judged win rates across different sampling temperatures.

## USE OF LARGE LANGUAGE MODELS

In addition to running LLMs in our experiments, we also employed them to assist and polish writing as well as baseline code writing. We take seriously the responsibility of using LLMs in research. All outputs were carefully monitored to avoid plagiarism, fact fabrication, or other forms of scientific misconduct, and we confirm that none of these issues are present in this paper.

