# OpenReview forum: "On the Significance of Softmax Geometry: Interpretability and Token Decoding"
_ICLR.cc/2026/Conference — Submitted to ICLR 2026_

### Official Review · Reviewer_NAoQ · 2025-10-28

**Soundness:** 2
**Presentation:** 1
**Contribution:** 3
**Rating:** 2
**Confidence:** 3

**Summary:**

This work studies the geometry underlying the softmax layer in large language models and argues that interpretability and retrieval should be done in the unembedding space equipped with the softmax-induced (whitened) inner product, rather than the standard Euclidean metric. The authors first show that training a sparse autoencoder (SAE) on whitened unembedding vectors yields latent features that are substantially more semantically coherent than those learned under Euclidean reconstruction, as judged by GPT-4. They then leverage the same geometry to build a token retrieval mechanism: instead of computing the full softmax over the entire vocabulary, they use the dual map $\phi(\lambda)$ to transform the model’s context embedding into unembedding space and retrieve likely next tokens via sparse SAE feature codes. Because the exact dual map requires a full softmax, the authors train a small MLP to approximate it. Finally, they show that this geometry-aware SAE retrieval recovers significantly more next-token probability mass than LSH baselines (even when LSH is performed in whitened space), and yields text generation quality comparable to standard greedy decoding.

**Strengths:**

The ideas presented in this work are novel and the inclusion of concrete examples strengthens the motivation for the proposed approach. The work effectively argues that operating in a geometry-aligned unembedding space can provide more interpretable latent features, and the qualitative illustrations (e.g., Figure 2) help justify this perspective. The observations made here also point toward a practical use case: leveraging these more interpretable representations for token retrieval during generation. Overall, the paper makes a clear case that geometry-aware representations can be both conceptually meaningful and potentially useful for improving decoding behavior.

**Weaknesses:**

- **W1)** A primary weakness of the paper is the clarity and organization of the exposition. The narrative is relatively strong up to the introduction of the dual map, but after this point the presentation becomes fragmented. The paper spends considerable space comparing Euclidean LSH, whitened LSH, and LSH under exact versus approximate dual mappings before introducing the SAE-based retrieval mechanism, which appears to be the method the authors ultimately intend to advocate. This emphasis on LSH variants feels out of proportion to their conceptual importance and reads more like an extended ablation study rather than a central contribution. As a result, the core idea—geometry-aware SAE-based retrieval using the approximate dual map—gets diluted and arrives later than it should. A clearer structure would have foregrounded the SAE-based retrieval method and used the LSH comparisons only as supporting evidence, rather than as a major narrative focus.

- **W2)** Additionally, I am not fully convinced that the decoding component warrants the level of emphasis it receives in the paper. Even for the SAE-based retrieval method, the central claimed benefit appears to be computational efficiency—specifically, avoiding the full vocabulary matrix multiplication during softmax. However, the paper does not provide any runtime measurements, latency comparisons, or FLOP estimates to demonstrate that the proposed retrieval actually results in meaningful speedups in practice. Without such evidence, it is difficult to evaluate whether this portion of the work represents a substantive contribution or merely a conceptual possibility. If the decoding mechanism is meant to be a practical improvement, the absence of performance data is a major gap, and if not, the extensive focus on retrieval seems disproportionate relative to the interpretability contributions.

- **W3)** Another concern is the limited scope of models and evaluation. The analysis is performed almost exclusively on Gemma models, and—as far as I can tell—there is no ablation on other model families to demonstrate whether the observed geometric structure or the retrieval mechanism generalizes. In terms of evaluation, the primary metrics are probability-mass coverage and GPT-4-based qualitative judgments. While these are reasonable given the interpretability focus, some quantitative language modeling metrics (e.g., perplexity before and after applying the proposed retrieval mechanism) would provide a more grounded assessment of the impact on model performance. In addition, the comparison to standard decoding is done using greedy decoding only, which is known to underperform relative to widely used sampling strategies. A more fair comparison would involve top-
𝑘
k or nucleus sampling for both the baseline and the proposed decoding scheme. Without these evaluations, it is difficult to assess whether the method preserves performance under commonly used inference settings.

- **W4)** (Minor) On a more minor note, the presentation is often difficult to follow, largely due to the extended focus on retrieval experiments. It is frequently unclear which combination of mapping strategy (exact or approximate dual), whitening space, and hashing method is being used in each figure, and reconstructing the experimental setup for each comparison requires significant effort. This confusion seems to stem from the fact that several of these retrieval experiments function more as supplementary analyses than as core contributions. A tighter structure might place results such as Table 1 and Figure 3 in the appendix (with only high-level summaries in the main text), reserving the main narrative for the geometry-aware SAE method and its interpretation. As written, the appendix is occupied primarily by text generation examples, while much of the methodological clarification that would help the reader is left implicit in the main body.

**Questions:**

**Q1)**: Could the authors clarify what specific practical advantage the proposed geometry-aware retrieval offers beyond conceptual interpretability? Is the main contribution intended to be interpretability, efficiency, or both?

---

> ### Author Response · Authors · 2025-11-21
>
> We chose not to write a rebuttal to this review because it appears to be LLM-generated: (1) the online detector indicates this (https://iclr.pangram.com/reviews?submission_number=21505), (2) it claims the paper is “super unclear” despite the other reviewers stating the opposite, (3) it requests experiments on sampling methods that are already included in Section 5.3, and (4) it contains repeated use of em dashes. For these reasons, we would like to flag this review to the AC.

---

> ### Comment · Reviewer_NAoQ · 2025-11-26
>
> Hello and sorry for replying late, I was going through my other reviews and this was the last one on the list. I will be long here and will not use any AI tools for rewriting. I also tend to write a lot, apologies but I prefer to clarify this misunderstanding.
>
> Yes I always write one paragraph at a time and ask an AI model to "make it nicer" or "make it more profesional". I can see why the flag would be set off. Perhaps this is my mistake and I apologise, will be more careful in the future. If the authors and AC believe that I didn't read the paper and my pointed out issues are invalid, I am happy to discuss the review and my mentioned weaknesses one by one with the AC. I can also provide pdf that I worked on while reviewing (I think I can show the last day modified) + my original prompts to the model if I can find them. I did truly read the paper (did have a hard time understanding it to be honest ) and all the main pointed out issues are ones that I wrote down while reviewing, albeit I could have presented them better.
>
>
> Here I will discuss Weakness 3, these are the main reasons I gave this paper a lower score in terms of soundness.
>
> I believes the issues I raise in my weakness 3 are valid, but I think I should have split them to be more clear, my bad.
>
> **Other decoding methods**: Firstly thank you for pointing out the top-p and top-k, I honestly missed the quantitative tables because it was after all the 10-15 pages of generated examples (which I can't really evaluate) and I thought the main body was referring to the **qualitiative examples only** for other decoding methods. My mistake in wording my concern and missing the table but please move that final figure 18 up for future revisions. I don't think an LLM would miss this detail, but I did !
>
> **Quantitatieve assesments:** I think my issues with quantitatieve assesment still hold. I am new to this literature and LLM as a judge seems at best a complementary evaluation tool to me. I thought seeing some results on **perplexity**, **Q/A** or perhaps **short summarizations** are a requirement for suggested methods (with F1 score or Rouge as metric). If the authors believe I am wrong on this, please let me know explicitly. Does this paper mainly focus on LLM-as-a-judge and text generation because the main focus is interpretability ? if so I can see an argument as evluating text generation is almsot not possible with other metrics.
>
> **Interpretability results on a single architecture only:** I should have been more clear regarding my issue with just using Gemma. I have seen results both in my own work and other papers where Gemma behaves differently than other models. An example would be this paper [1], where at Fig 3, you can see Gemma have different behaviour for its geometry across scales comapred to the other models. So I asked for additional models to confirm the result. I think that would help improve the soundness of this work. I understand that the time is limited and I replied late, but if the authors can provide an example, I would appreciate it and remove that from my concerns for the paper. I believe **Llama-2 was originally used by Park et al 2024-2025** and should be good candidate. Just the quantitative results on LLM-as-a-Judge would be fine enough even.
>
> [1]  Lee A, Weber M, Vi´egas F,  Wattenberg M:  Shared Global and Local Geometry of Language Model Embeddings

---

> ### Comment · Reviewer_NAoQ · 2025-11-26
>
> regarding the runtime concerns, if the authors are providing a method for SAE for approximating the top-k tokens more efficiently, then they need to show more efficiency analysis. I can see the arguments regarding fig 4 (how the selected subset of tokens is on the order of 1-4K on average, much smaller than full vocab size) and I understand the argument provided on scailibility complextiy in line 284. But there still need to be a comparison between the different decoding techniques. Passing the embedding through an auto encoder and then the learnt dual map network are additional computation costs that aren't accounted in this paper's analysis. I think this may stem from my lack of understand for the logit computations wich this model skips. If its unfair of me to ask this, please explain and I will revoke this point.
>
> note: I have read the response to reviewer B91w and I don't think this explain my question.

---

> > ### Comment · Reviewer_NAoQ · 2025-11-26
> >
> > Finally, regarding readability and paper format (Weaknesses 1 and 4), I think the main method of the paper is obfuscated by the extensive explanations around the choice of hash functions and the duality mapping. While I understand that the explanations and the additional experiments, such as the hashing comparison in Figure 3 or the analysis of exact dual functions vs. trained neural networks, are necessary, they should be positioned more as ablations rather than as the main focus. A single visual illustrating the finalized decoding strategy would greatly clarify the main idea, and providing more quantitative results with additional models could further increase the reader’s confidence in the proposed method.
> >
> > I personally found it difficult to follow the main argument, to understand some of the terminology, and identify what the final method was, which is why I voiced my concerns. I also believe that this lack of clarity is echoed by other reviewers, even if they ultimately gave higher presentation scores.

---

> > > ### Comment · Reviewer_NAoQ · 2025-11-26
> > >
> > > Finally, apologies for the long winded explanation but I hope this clarifies the confusion. I would appreciate if the authors can respond to some of my concerns in the limited time, I would be happy to raise my score as the main points are addressed. I believe this work is novel and interesting, hence the 3 on contribution, but I also believe it lacks required content and would benefit from a revision. If needed, I am happy to continue this discussion with the AC to further prove the validity of this review.

---

### Official Review · Reviewer_SAcD · 2025-10-31

**Soundness:** 3
**Presentation:** 3
**Contribution:** 3
**Rating:** 6
**Confidence:** 4

**Summary:**

The paper builds on the findings of Park et al. (2024), which argued that Euclidean metrics are not well-suited for analyzing language model embeddings and introduced the causal inner product, an inner product in the whitened coordinate space of embeddings, derived from and motivated by the softmax formulation. This work investigates the significance of this perspective on embedding geometry for both interpretability and efficiency.

First, the authors train sparse autoencoders (SAEs) on the unembedding vectors using the causal inner product, showing that this yields more interpretable hidden representations compared to standard Euclidean-based SAEs used in prior work. They demonstrate this by listing the tokens with the highest activation values for each SAE dimension under both metrics, finding that the token sets are more semantically coherent when analyzed under the softmax-geometry view.

The paper then leverages this more interpretable SAE representation to approximate top-k next-token retrieval without computing logits over the full vocabulary. In this setup, the SAE encoding functions as a form of locality-sensitive hashing that proposes a small candidate set of likely tokens for a given context embedding, from which the nearest tokens are then selected. The authors compare their approach against explicit hashing and other sampling methods, showing that the causal-metric SAE achieves better next-token probability mass coverage or produces higher-quality candidate sets when evaluated using GPT as a judge.

**Strengths:**

The paper is well-written and clearly motivated. The authors present a logical progression of ideas leading to their proposed methodology, and the step-by-step explanations make the paper easy to follow.

While the new geometry perspective is not new and builds on prior work, the paper is a nice follow-up on the benefits this perspective can offer. The experimental evidence is also generally well-aligned with the paper’s claims.

**Weaknesses:**

While the paper is generally well-written and clear, certain implementation details and results could be discussed more thoroughly. I elaborate on some of these issues in the Questions section.

**Questions:**

1. **Sec 3**: I have a few questions regarding terminology and implementation details of the SAE experiment:

    (i) Could you clarify what the listed tokens represent for each activation in Figures S1–S9? Are these the tokens that achieve the highest activation values at a specific feature dimension (regardless of which features are most active for them), or are they the tokens whose top-activated feature corresponds to that specific dimension?

    (ii) In the caption of Figure 1, what exactly is meant by “SAE features" and their relevance? Does this refer to the relevance of the token lists shown in Figures S1–S9 to the target word?

    (iii) Related to (ii), in the prompt template shown in Figure S10, do *response_a* and *response_b* refer to these token lists for each feature?

2. **Fig 3**: Do you have any intuition for why increasing the number of hash functions and hash tables appears to have opposite effects on the probability mass coverage?

3. **Sec 4.3**:

    (i) In Step 3 of the top-k token approximation, are the selected tokens those that share the exact same set of non-zero entries, or are they chosen based on having overlapping active encoding dimensions with the context embedding? Related to this, in Fig 4, what defines the size of the candidate set? Is it the size of the tokens that have the same set of non-zero entries?

    (ii) Regarding the reported time complexity, does it account for the cost of searching over tokens that share the same non-zero entries, or only for computing the softmax over the reduced candidate set? Additionally, this approach likely introduces some extra memory cost (for example, storing the precomputed dictionary) as well as computational overhead for approximating the dual map and training the SAE. Could you also report/comment on these costs?

5. **Fig 6**: How should this figure be interpreted? If “win” indicates cases where the SAE produces better next-token candidates, SAE would be better when the dark purple bars are larger than the orange ones, which does not appear to be the case. Could you clarify why you conclude that the performance of greedy decoding and your approximation approach are on par?

6. **Minor**: The axis font sizes in figures are very small and difficult to read.

---

> ### Author Response · Authors · 2025-11-21
>
> >  1 (i) Could you clarify what the listed tokens represent for each activation in Figures S1–S9? Are these the tokens that achieve the highest activation values at a specific feature dimension (regardless of which features are most active for them), or are they the tokens whose top-activated feature corresponds to that specific dimension?
>
> Sorry for the confusion, but you are right. These are the tokens with the highest activation values on that feature dimension, independent of which features are most active for those tokens overall.
>
> > 1 (ii) In the caption of Figure 1, what exactly is meant by “SAE features" and their relevance? Does this refer to the relevance of the token lists shown in Figures S1–S9 to the target word?
>
> Yes, we use tokens as a way to capture the semantic meaning of SAE features.
>
> > 3 (iii) Related to (ii), in the prompt template shown in Figure S10, do response_a and response_b refer to these token lists for each feature?
>
> Yes, exactly, we just put the structured outputs directly into the prompts.
>
>
> > 2. Fig 3: Do you have any intuition for why increasing the number of hash functions and hash tables appears to have opposite effects on the probability mass coverage?
>
> Yes, increasing the number of hash functions makes the partition of the space more granular. This can be harmful when the underlying geometry does not align. For example, in Figure 2, where embeddings and unembeddings live in separate spaces, using only one hash function causes them to collide every time, effectively reducing the method to standard decoding. Using multiple hash functions lowers the collision rate, which can lead to less shared information being captured. Separately, increasing the number of hash tables introduces more statistically independent hash functions, which can generally improve performance regardless.
>
> > 3 (i) In Step 3 of the top-k token approximation, are the selected tokens those that share the exact same set of non-zero entries, or are they chosen based on having overlapping active encoding dimensions with the context embedding? Related to this, in Fig 4, what defines the size of the candidate set? Is it the size of the tokens that have the same set of non-zero entries?
>
> Good question. The tokens are selected based on overlapping active encoding dimensions. In other words, the candidate set consists of any tokens that share active encoding dimensions with the target token.
>
> > 3(ii) Regarding the reported time complexity, does it account for the cost of searching over tokens that share the same non-zero entries, or only for computing the softmax over the reduced candidate set? Additionally, this approach likely introduces some extra memory cost (for example, storing the precomputed dictionary) as well as computational overhead for approximating the dual map and training the SAE. Could you also report/comment on these costs?
>
> The complexity of searching over tokens is just $O(\tilde{V})$ where $\tilde{V}$ is the effective vocabulary size. This is because we store the mapping between tokens and their encoding dimensions using hash maps, so at inference time we simply retrieve the relevant tokens from these maps. This does introduce additional memory overhead, but it is unavoidable. The cost of computing the dual map is constant with respect to vocabulary size, and the SAE only needs to be trained once before inference.
>
> > 4. Fig 6: How should this figure be interpreted? If “win” indicates cases where the SAE produces better next-token candidates, SAE would be better when the dark purple bars are larger than the orange ones, which does not appear to be the case. Could you clarify why you conclude that the performance of greedy decoding and your approximation approach are on par?
>
> Because our method is only an approximation to standard decoding, it can never strictly outperform it. Therefore, reporting only win or lose percentages would be misleading. We include “ties” in our win-rate graph to indicate cases where our method matches standard decoding while relying only on top-k token retrievals.

---

### Official Review · Reviewer_B91w · 2025-11-01

**Soundness:** 3
**Presentation:** 3
**Contribution:** 1
**Rating:** 2
**Confidence:** 4

**Summary:**

This paper studies the unembedding process in LLMs. By training SAEs they find applying a whitening transformation improves the interpretability of features in the embedding and unembedding vectors in line with previous work. The authors then introduce an approximation to the softmax decoding procedure leveraging SAE based locality sensitive hashing to reduce the complexity of the unembedding process while retaining performance.

**Strengths:**

This paper is well written and clearly articulates the work. Formalisations are clear and concise, making it straightforward for the reader to follow. The problem studied is of some interest given the general interest in accelerating decoding in LLMs.

**Weaknesses:**

In section 3 the authors claim SAEs trained with the causal inner product produce more interpretable features. The evidence for this is qualitative, or based on showing the features to an LLM and having it judge. Neither of these are compelling sources of evidence. The qualitative analysis points to the appendix for an example relating to the word puppy, however this appears to be cherry-picked given the example on the next page (for queen) shows reasonable interpretable cluster for both methods. The LLM as judge results are a bit concerning - given this is a task about discerning how well grouped semantic concepts are it seems possible this may in fact reflect how well aligned the results are with the semantic concepts of the judge model. Neither of these evaluations seem particularly quantitive.

Looking at the performance of the SAE decoding (figure 6) it is somewhat concerning the win rate for your procedure is lower than its win rate in every condition. To make claims about parity between approaches here some statistical testing may be helpful.

For the decoding results, the authors state that their approximation "could offer significant computational benefits" (line 484). It seems odd that the decoding results do not quantify the computational complexity of their approximation or empirically compare its efficiency with existing methods. Without comparing to a baseline it is difficult to evaluate the efficiency of this approximation.

Without this approach being clearly better in terms of win rate, or demonstrably faster, the novelty here appears limited. In terms of presentation, the paper would benefit form a proofread (e.g. line 356 misspelled autoencoder) and the plots in figure 2, 3, and 5 are difficult to read (the text size is quite small!).

**Questions:**

1. What is the quantitative evidence that your approach is faster, but equally as good as existing decoding techniques.

2. Does applying a whitening transformation improve standard decoding?

3. Have you tried this approach on any other model families (even if they're small models) to ensure it generalises?

---

> ### Author Response · Authors · 2025-11-21
>
> > What is the quantitative evidence that your approach is faster, but equally as good as existing decoding techniques.
>
> The main point of the paper is that our token retrieval method can enable **scalable** decoding. The gain here is simply approximate top-k vs exact top-k. Generically, relaxing the exact nearest neighbors to approximate yields large computational complexity gains (e.g., that’s basically the idea behind LSH). See discussion in section 4.3. Empirically, Figure 4 shows we can skip computing logits for the vast majority (more than 98%) of the vocabulary, which we consider to be quantitative evidence.
>
> The problem that is salient in this paper is that these efficiency gains are only useful if the approximation is actually good! The result illustrated by the experiments (table 1, figure 5) is that accounting for the non-Euclidean geometry greatly improves the approximation quality and figure 6 shows this minimally affects the text generation quality. We want to emphasize that our method is an approximation to the standard decoding. Therefore, we only expect the performance to be on par, which is shown by the win rate plots.
>
> We also note that the particular task we’re considering here is mainly chosen because the geometry is understood exactly. The point is that the non-euclidean geometry is substantive. It is also true that this particular experiment may have some significant practical implications—e.g., in edge models such as Gemma 3n the vocabulary computation is a significant part of the workload; additionally the approach here enables ultra-large vocabularies such as those occurring when using dynamic vocabulary construction—but this is not our focus, and we leave it to future work.
>
> > Does applying a whitening transformation improve standard decoding?
>
> It does help a little. This is shown in Table 1. The experiments are in the context of top-K token retrievals because standard decoding just computes the whole probability vectors.
>
> > Evaluation on SAE
>
> We are sympathetic to concerns about potential biases in interpretability evaluations, and this is precisely why we provide multiple forms of evidence. Our qualitative assessments include concrete examples, and our quantitative evaluation uses GPT-4o as a judge to compare our method against the baseline at scale. We also include notebooks in the supplementary materials so reviewers can experiment with tokens of their choice. Although none of these evaluations are perfect, they consistently point to the same conclusion: the whitened SAE yields better features, and we believe these findings will be useful to the interpretability community. In general, there is no perfect interpretability metric, and coming up with good ones is an active research area in its own right.

---

### Official Review · Reviewer_6quy · 2025-11-01

**Soundness:** 1
**Presentation:** 3
**Contribution:** 1
**Rating:** 2
**Confidence:** 3

**Summary:**

This manuscript proposes to empirically evaluate the relevance of a similarity measure borrowed from Park et al (2024, 2025) as an alternative to the usual Euclidean distance or cosine distance, focusing on two tasks: learning interpretable features through sparse autoencoders and finding the $k$ most probable next tokens given a context embedding. The paper claims that using such a measure has a "dramatic" effect on the performance on both tasks.

**Strengths:**

- Studying LLMs from a geometrical perspective is a promising orientation in the field
- The idea of addressing SAE as hash functions in the context of a study of LLM geometry is an interesting one
- The manuscript is well-written

**Weaknesses:**

- Unclear presentation of technical details. In particular, LLMs is not a technical term, and it's not clear what specific architecture (LSTMs, transformers, SSMs) is being described, for instance, when it is claimed that $x$ is a context sentence that "the LLM" encodes as a vector. Another example, the vectors $\lambda_x, \gamma_y$ are said to "not live in the same space", while they both belong to $\mathbb{R}^d$ (otherwise they couldn't be multiplied in the first place). One can guess that the manuscript means something different with "space" here (maybe that's what's intended with $\Lambda$ and $\Gamma$ being isomorphic to $\mathbb{R}^d$?), but then it's not clear what these spaces are (the image of some map?) and how multiplication (say, a dot product) is defined over both. Figure 2 seems to give an intuition by showing a plot of PCA, but it is not clear how the differently colored dots rigorously (i.e., formally) define a space, let alone a "subspace", as claimed in the legend. It is also unclear why it is claimed that the fact that "biproducts" (dot products?) are invariant under some invertible linear transformation, while dot products between "embeddings" and between "unembeddings" are not, means that Euclidean geometry is not "privileged" by softmax distribution (what does "privilege" mean here, formally?). All these concepts could accept a rigorous definition and treatment, but they are treated here in a highly intuitive and sloppy way, which makes it difficult to assess the extent to which what is claimed has any solid basis.

- Uncritical adoption of an alternative metric. The manuscript constantly refers to "the correct geometry" ("natural", "better adapted to the structure of LLMs") without providing any argument or explanation of why this is correct, or even correct with respect to what, and uncritically referring to and relying on Park et al (2024), where the measure is proposed and used for specific needs. In particular, referring to that line of work, it is said that what justifies the proposed metric is that it "aligns well with the structure of semantics as understood by humans", "semantic structure", and "human intuition about semantic similarity". I, personally, don't have that reading of that work, where I can't find anything related to human alignment, analysis of human intuition about semantic similarity, or serious engagement with semantic structure (other than orthogonality of concepts), and I have serious reservations with respect to the claims made in Park et al (2025) concerning semantic "structure". But this is not the place to judge that work. The point is that the present manuscript adopts them uncritically, probably extending the claims beyond what that work claimed or could deliver, to justify the work performed here without further analysis or elaboration. If these claims are set aside as unsubstantiated, the paper’s scope becomes substantially limited: it reduces to applying a single measure to two tasks. Even if the task evaluations were sound (see the next point), the contribution is, in my view, too incremental to warrant publication in this venue.

- The evaluation of tasks is invalid. The manuscript claims that "training sparse autoencoders with a causal inner product yields more interpretable features". Yet, it is known that "more interpretable" is not a well-defined notion. To address this, the manuscript relies on subjective assessments of cherry-picked examples; for instance, representations of "puppy" are described as displaying a clear "dog" feature or as "appearing nonsensical". As an alternative "quantitative evaluation", the manuscript appeals to GPT-4o as a judge. Likewise, the manuscript evaluates the outputs of the SAE method as "fluent and human-like" by asking GPT-4o to "please act as an impartial judge and evaluate the quality of snippets of paragraph generated by two language models". All of which I consider radically unscientific. Incidentally, on the second task, transformations are not computed through the exact dual map, but through another map learned through an MLP, so in the end, it is even unclear what is it that is being evaluated.

**Questions:**

- What is your justification for considering it as a rigorous scientific method to prompt chatGPT to please be an impartial judge?
- In which sense is the measure proposed by Park et al (2024) "the correct geometry" (or "natural", or "better adapted to the structure of LLMs"?
- What is your evidence for claiming that such a measure aligns with human intuition?
- Can you provide a rigorous (formal) definition of what you mean by "space" and "subspace" and a rigorous account of how "embeddings" and "unembeddings" relate to it?

---

> ### Author Response · Authors · 2025-11-21
>
> >  On SAE evaluation
>
> We agree with the reviewer that interpretability is not a well-defined concept. As with many aspects of LLMs, evaluation is challenging but not impossible. To that end, we provide qualitative assessments with examples, and runnable notebooks in the supplementary materials so reviewers can experiment with words of their choice. For quantitative evaluation, we use GPT-4o as a judge to compare our method against the baseline at a larger scale. While none of these constitute definitive scientific proofs, they offer practical and informative ways for evaluating phenomena that are inherently difficult to measure.
>
> > On Dual map evaluation
>
> We include experiments using the exact dual map (see Table 1). However, computing the exact dual map is not practical in general and defeats the purpose of the paper, so we also provide an MLP-approximated version to demonstrate the practical benefits of our approach.
>
> > On contributions
>
> This paper does more than apply a single measure to two tasks. One of the central contributions is the introduction of a dual map between the embedding and unembedding spaces which is not present in Park et al. (2024). While the causal inner product offers one way to define an inner product within the unembedding space, the dual map enables embeddings to be transformed into that space, allowing us to use its inner product to measure a notion of similarity between embeddings and unembeddings.
>
> > Can you provide a rigorous (formal) definition of what you mean by "space" and "subspace" and a rigorous account of how "embeddings" and "unembeddings" relate to it?
>
> Unembeddings live in a linear space $\mathbb{R}^d$. Embeddings can be viewed as linear functionals that map elements (unembeddings) in this space to $\mathbb{R}$.
> In general, any finite-dimensional vector space is linearly isomorphic to its dual, but there is no canonical way to identify the two. Once the space is endowed with an inner product, the Riesz representation theorem gives a way to map functionals to vectors. However, this identification depends entirely on the choice of inner product. There is no unique or “correct’’ inner product on $\mathbb{R}^d$: multiple valid ones can be defined.
>
> This relates to the last softmax layer and exists in almost all language models.

---

### Meta-Review · Area_Chair_mhKh · 2025-12-18

**Summary:**

This paper is a follow-up on earlier work by Park et al., which argued that the Euclidean geometry is ill-matched for LLMs and gave a similarity metric based on the geometry of the softmax. The manuscript considers learning interpretable features via SAEs and retrieval of top-k most probable tokens, giving some evidence of the advantages of the proposed geometry.

It has been pointed out in the reviews that the angle considered in this work (geometric perspective based on the softmax) is interesting, and the experimental evidence is aligned with the main message. However, the evidence itself has been deemed insufficient by multiple reviewers (6quy, B19w, NAoQ): the findings are mostly qualitative and using an LLM as a judge may introduce an evaluation bias. The rebuttal unfortunately did not address these issues in a satisfactory way and therefore I am recommending a rejection at this stage.

Having said that, I do think the paper has potential and I would encourage the authors to provide more convincing evaluation metrics in the context of a re-submission.

**Reviewer Concerns:**

Reviewer SAcD (who was already positive) posed a number of detailed questions in the review which were answered in a satisfactory way in the rebuttal. In contrast, I don't find that the most critical concerns coming from the other reviewers were addressed in the rebuttal, also because they would require a substantial revision. In particular, I find that the main concern raised by the reviewers and that remains outstanding is the qualitative nature of the findings and the usage of an LLM as a judge. Another important shared issue is that the computational benefits of the method are not really quantified.

The authors opted not to respond to Reviewer NAoQ claiming that the review was generated by an LLM. Said reviewer provided a detailed answer essentially saying that an LLM was used only for polishing the writing and not for generating content. I find this plausible: some of the concerns are actually the same as those appearing in other reviews (using an LLM as a judge, concerns on runtime).

**Reviewer Scores:**

While a full discussion may have led to slightly increased scores, upon reading the reviews, the rebuttal and the paper, I do not think that the consensus would have changed towards a recommendation to accept the paper in its current form.

---

### Decision · Program_Chairs · 2026-01-26

Reject